# TACKLING UNDERESTIMATION BIAS IN SUCCESSOR FEATURES BY DISTRIBUTIONAL REINFORCEMENT LEARNING

## ABSTRACT

The framework of successor features (SFs) and generalized policy improvement (GPI) yields the potential to achieve zero-shot transfer in reinforcement learning (RL) among different tasks. However, GPI always suffers from inaccurate value function approximation in practice, resulting in a "zero-shot" somewhat fantastical. This paper focuses on comprehending the underlying causes of inaccurate SFs and presents a methodology for improving their accuracy. Our contributions encompass four key aspects: (i) we theoretically study the underestimation phenomenon in SF&GPI; (ii) we introduce distributional RL into SF&GPI, and demonstrate its effectiveness in relieving such underestimation; (iii) we show that distributional SFs (DSFs) is provided with a lower generalization bound than original SFs; (iv) we put forward that the performance of SFs-based algorithms can be enhanced by incorporating DSFs. Furthermore, we verify the quality of employing DSFs on the platform of multi-objective RL (MORL). Simulation study demonstrates the superiority of our concept in addressing underestimation challenges.

## 1 INTRODUCTION

Reinforcement learning (RL) (Sutton & Barto, 2018; Puterman, 2014) solves sequential decision-making problems via a trial-and-error process interacting with the environment, which achieves groundbreaking success in game playing (Schrittwieser et al., 2020), robotics (Kumar et al., 2021), autonomous driving (Sallab et al., 2017).

Model-free RL algorithms, a cornerstone in the field of RL, consistently excel at solving complex problems and achieving state-of-the-art results. However, many of these algorithms often suffer from overestimation, which is commonly caused by inaccurate function estimation or maximization operations, as executed by Q-learning and its variants (Fujimoto et al., 2018; Pan et al., 2020b; Duan et al., 2022). To address this overestimation, (Fujimoto et al., 2018; Lan et al., 2020) employ minimization operation but it brings another risk of underestimation. Consequently, those model-free RL methods will degenerate to sub-optimal policy (Duan et al., 2022). Overcoming or even relieving the chronic overestimation/underestimation bias has great significance in cost saving, and has aroused great interest of scholars (Pan et al., 2020b;a; Duan et al., 2022).

**Will this unpleasant thing happen in the transfer reinforcement learning (TRL) algorithms?** The scholarly exploration that overlooks this issue presumably violates the original intention of TRL: Low sample complexity. Explosively, we take an impressive TRL method - successor features (SFs) (Barreto et al., 2017; 2018; Carvalho et al., 2023) as an example, to study the underlying overestimation/underestimation bias. The SFs framework is promising as it facilitates seamless task transfer with generalized policy improvement (GPI) regardless of temporal order, seamlessly integrating with RL.

In this paper, we theoretically expose the underestimation in SF&GPI by exploring the post-update of the estimated parameter and the true parameter. To mitigate the underestimation bias, we incorporate the idea of distributional RL (Sobel, 1982; White, 1988; Bellemare et al., 2017) into SF&GPI, and define distributional SFs (DSFs) and distributional GPI (DGPI), respectively. Distributional RL captures the randomness and brings more information into value function approximation via model-

ing distribution over return (Bellemare et al., 2017). From the view of such an approximation mechanism, we theoretically prove that the underestimation is mitigated by distributional SF&GPI, and demonstrate that DSFs are provided with access to a lower generalization bound than original SFs. They enrich our concepts mutually. Finally, we put forward that the performance of SFs-based algorithms can be enhanced by incorporating DSFs. Furthermore, we employ the SF-based extension of the optimistic linear support (SFOLS) algorithm (Alegre et al., 2022b) and the worst case policy iteration (WCPI) algorithm (Zahavy et al., 2021) as the platform to verify the quality of involving DSFs. The novel DSFs-based algorithms as the control groups are named by the Risk-sensitive Distributional SFs with Optimistic Linear Support (RDSFOLS) algorithm and the Distributional Generalized Policy Improvement-Worst Case Policy Iteration (DGPI-WCPI) algorithm, respectively. Extensive quantitative evaluations support our analysis.

## 2 BACKGROUND

We consider an Markov decision process (MDP) (Sutton & Barto, 2018; Puterman, 2014) defined by a tuple $M = (\mathcal{S}, \mathcal{A}, p, R, \mu, \gamma)$. $\mathcal{S}$ and $\mathcal{A}$ represent the state space and the finite action space, $s' \sim p(\cdot|s, a)$ describes the transition dynamics, $R : \mathcal{S} \times \mathcal{A} \times \mathcal{S} \to \mathbb{R}$ is the reward function, $\mu$ is an initial state distribution and $\gamma \in [0, 1)$ is the discounting factor. The action-value function of the policy $\pi : \mathcal{S} \to \mathcal{A}$ is given by $Q^\pi(s, a) \equiv \mathbb{E}^\pi[G_t|S_t = s, A_t = a]$, where $\mathbb{E}^\pi[\cdot]$ denotes the expectation over trajectories induced by $\pi$, and $G_t = \sum_{i=0}^{\infty} \gamma^i R(S_{t+i}, A_{t+i}, S_{t+i+1})$. In the following, we introduce preliminary background about SF&GPI and MORL. Details for distributional RL can be found in Appendix B.

### 2.1 SUCCESSOR FEATURES AND GENERALIZED POLICY IMPROVEMENT

SF&GPI (Barreto et al., 2017; 2018) is a powerful technique for transfer in RL, which fully utilizes the policies from prior tasks to identify a policy for a novel task. Thus we consider the linearly-expressible reward function $\mathbb{E}[R(s, a, s')] = r_{\boldsymbol{w}}(s, a, s') = \boldsymbol{\phi}(s, a, s')^\top \boldsymbol{w}$, where $\boldsymbol{\phi}(s, a, s') \in \mathbb{R}^d$ are reward features and $\boldsymbol{w} \in \mathbb{R}^d$ are weights. Let $\mathcal{M}^\phi = \{(\mathcal{S}, \mathcal{A}, p, r_{\boldsymbol{w}}, \mu, \gamma)|r_{\boldsymbol{w}}(s, a, s') = \boldsymbol{\phi}(s, a, s')^\top \boldsymbol{w}\}$ be the set of MDPs induced by $\boldsymbol{\phi}$ through all linearly-expressible reward functions. The key insight of SFs is to decompose the action-value function of policy $\pi$ on task $\boldsymbol{w}$:

$$Q^\pi_{\boldsymbol{w}}(s, a) = \mathbb{E}^\pi[\sum_{i=0}^{\infty} \gamma^i \boldsymbol{\phi}_{t+i}|S_t = s, A_t = a]^\top \boldsymbol{w} \equiv \boldsymbol{\psi}^\pi(s, a)^\top \boldsymbol{w}. \tag{1}$$

Analogous to the approximation of the Q-function, we employ the parameter $\boldsymbol{\theta}$ to approximate SFs $\boldsymbol{\psi}^\pi(s, a; \boldsymbol{\theta})$. Note that SFs can be learned through any conventional RL method (Szepesvári, 2022).

The property of SFs allows for the reuse of SFs across a set of policies, thereby accelerating policy updates. Assume the agent has learned the SFs $\boldsymbol{\psi}^{\pi_j}(s, a; \boldsymbol{\theta}_j)$ of policies $\Pi = \{\pi_j\}_{j=1}^n$. For a new task $\boldsymbol{w}_{n+1}$, it is practicable to evaluate all policies $\pi_j \in \Pi$ via generalized policy evaluation (GPE), i.e., $Q^{\pi_j}_{\boldsymbol{w}_{n+1}}(s, a; \boldsymbol{\theta}_j) = \boldsymbol{\psi}^{\pi_j}(s, a; \boldsymbol{\theta}_j)^\top \boldsymbol{w}_{n+1}$. Next, we apply generalized policy improvement (GPI) to obtain a new policy

$$\pi_{n+1}(s) = \arg\max_a \boldsymbol{\psi}^{\pi_i}(s, a; \boldsymbol{\theta}_i)^\top \boldsymbol{w}_{n+1}, \ i = \arg\max_{j \in [n]} \boldsymbol{\psi}^{\pi_j}(s, b; \boldsymbol{\theta}_j)^\top \boldsymbol{w}_{n+1} \tag{2}$$

where $[n] = \{1, \dots, n\}$ and $\boldsymbol{\psi}^{\pi_j}(s, b; \boldsymbol{\theta}_j)^\top \boldsymbol{w}_{n+1}$ be the action-value function of policy $\pi_j$ when executed in $M_{n+1} \in \mathcal{M}^\phi$. The GPI theorem (Barreto et al., 2017) states that $\pi$ is no worse than all other training policies, i.e.,

$$Q^\pi_{\boldsymbol{w}_{n+1}}(s, a) \geq \max_i Q^{\pi_i}_{\boldsymbol{w}_{n+1}}(s, a), \text{ for all } (s, a) \in \mathcal{S} \times \mathcal{A}. \tag{3}$$

However, it always acts according to the lower bound on the action-value, which tends to underestimate the true value (Hunt et al., 2019). In this paper, to investigate the mechanism of this phenomenon, we assume $\pi_\star$ be the optimal policy in new task $\boldsymbol{w}_{n+1}$.

### 2.2 BRIDGING SUCCESSOR FEATURES AND MULTI-OBJECTIVE RL

MORL aims to tackle multiple possibly conflicting objectives, which can be modeled as a multi-objective MDP (MOMDP) $\mathcal{M} = (\mathcal{S}, \mathcal{A}, p, \boldsymbol{R}, \mu, \gamma)$. Differing from the regular MDP $M$ with its scalar reward $R$, MOMDP's reward function $\boldsymbol{R} : \mathcal{S} \times \mathcal{A} \times \mathcal{S} \to \mathbb{R}^m$ is with $m$ objectives.

With the aid of SFs, an MOMDP with $m = d$ objectives can be constructed by $\boldsymbol{R}(s, a, s') = \boldsymbol{\phi}(s, a, s')$ (Alegre et al., 2022b). Consequently, the multi-objective action-value function evolves into

$$\boldsymbol{q}^\pi(s, a) = \mathbb{E}^\pi \left[ \sum_{i=0}^\infty \gamma^i \boldsymbol{R}_{t+i} | S_t = s, A_t = a \right] = \mathbb{E}^\pi \left[ \sum_{i=0}^\infty \gamma^i \boldsymbol{\phi}_{t+i} | S_t = s, A_t = a \right] = \boldsymbol{\psi}^\pi(s, a).$$

Therefore, any algorithms capable of learning multi-objective action-value $\boldsymbol{q}^\pi(s, a)$ of a corresponding MOMDP can be used to learn the SFs $\boldsymbol{\psi}^\pi(s, a)$, and vice-versa. Further, the multi-objective value vector $\boldsymbol{v}^\pi$, defined as $\boldsymbol{v}^\pi := \mathbb{E}_{S_0 \sim \mu}[\boldsymbol{q}^\pi(S_0, \pi(S_0))]$ under the initial state distribution $\mu$, is equal to the expected SF vector $\boldsymbol{\psi}^\pi := \mathbb{E}_{S_0 \sim \mu}[\boldsymbol{\psi}^\pi(S_0, \pi(S_0))]$. The solution to an MOMDP is a set of all policies such that $\boldsymbol{v}^\pi$ is in the Pareto frontier.

Under the relative weight $\boldsymbol{w}$ describing the importance of $m$ objectives, let a user utility function (or scalarization function) be a mapping from the multi-objective value $\boldsymbol{v}^\pi$ onto a scalar value. The utility function is often linear, i.e., $u(\boldsymbol{v}^\pi, \boldsymbol{w}) = (\boldsymbol{v}^\pi)^\top \boldsymbol{w}$. According to (Roijers et al., 2013), we can define a convex coverage set (CCS) of the Pareto frontier. Given a constant $\boldsymbol{w}$, the MOMDP can be decomposed by an MDP with the reward function $r_{\boldsymbol{w}}(s, a, s') = \boldsymbol{R}(s, a, s')^\top \boldsymbol{w}$. Further, we can define the CCS by replacing each occurrence of $\boldsymbol{v}^\pi \in \mathcal{F}$ with its corresponding $\boldsymbol{\psi}^\pi$:

$$\mathrm{CCS} \equiv \{ \boldsymbol{v}^\pi \in \mathcal{F} \mid \exists \boldsymbol{w} \text{ s.t. } \forall \boldsymbol{v}^{\pi'} \in \mathcal{F}, u(\boldsymbol{v}^\pi, \boldsymbol{w}) \geq u(\boldsymbol{v}^{\pi'}, \boldsymbol{w}) \}$$
$$= \{ \boldsymbol{\psi}^\pi \mid \exists \boldsymbol{w} \text{ s.t. } \forall \boldsymbol{\psi}^{\pi'}, (\boldsymbol{\psi}^\pi)^\top \boldsymbol{w} \geq (\boldsymbol{\psi}^{\pi'})^\top \boldsymbol{w} \}$$
$$= \{ \boldsymbol{\psi}^\pi \mid \exists \boldsymbol{w} \text{ s.t. } \forall \pi', v_{\boldsymbol{w}}^\pi \geq v_{\boldsymbol{w}}^{\pi'} \},$$

where $v_{\boldsymbol{w}}^\pi = (\boldsymbol{\psi}^\pi)^\top \boldsymbol{w}$. Thus, we can exploit MORL algorithms tailored to construct CCS's to solve all tasks in $\mathcal{M}^\phi$. Through GPI, we can construct the set of optimal policies to solve all tasks in $\mathcal{M}^\phi$, which is equal to CCS solving the corresponding MOMDP.

## 3 UNDERESTIMATION BIAS

In Section 3.1, we provide a detailed analysis from a theoretical standpoint to expose the mystery of underestimation in the SF&GPI framework. Next, we show that DSF&DGPI not only mitigates underestimation but also narrows the generalization bound to some extent in Section 3.2. Finally, Section 3.3 extends the result to the scenario of MORL.

### 3.1 UNDERESTIMATION IN SFS TRANSFER FRAMEWORK

For a new task $\boldsymbol{w}_{n+1}$ (a constant weight), based on Eq. (1), the SFs-estimate $\boldsymbol{\psi}^{\pi_{n+1}}(s', a'; \boldsymbol{\theta}_{n+1})$ can be updated by minimizing the loss $(y - \boldsymbol{\psi}^{\pi_{n+1}}(s', a'; \boldsymbol{\theta}_i)^\top \boldsymbol{w}_{n+1})^2/2$:

$$\boldsymbol{\theta}_{n+1}^{\mathrm{new}} = \boldsymbol{\theta}_{n+1} - \beta\gamma(y - \boldsymbol{\psi}^{\pi_{n+1}}(s', a'; \boldsymbol{\theta}_i)^\top \boldsymbol{w}_{n+1}) \nabla_{\boldsymbol{\theta}_{n+1}} \mathbb{E}_{s'} \left[ \boldsymbol{\psi}^{\pi_{n+1}}(s', a'; \boldsymbol{\theta}_{n+1}) \right]^\top \boldsymbol{w}_{n+1}, \quad (4)$$

where $y = \mathbb{E} \left[ \boldsymbol{\phi}(s, a)^\top \boldsymbol{w}_{n+1} \right] + \gamma \mathbb{E}_{s'} \left[ \boldsymbol{\psi}^{\pi_{n+1}}(s', a'; \boldsymbol{\theta}_{n+1}) \right]^\top \boldsymbol{w}_{n+1}$ is the greedy target value and $\beta$ is the learning rate. Note that the target action $a'$ is chosen by $\arg\max_b \boldsymbol{\psi}^{\pi_{n+1}}(s, b; \boldsymbol{\theta}_{n+1})^\top \boldsymbol{w}_{n+1}$ and the current action is derived by the GPI policy in Eq. (2).

Let $\boldsymbol{\theta}_{n+1}^{\mathrm{true}}$ represent the post-update parameters derived from the true current value $\boldsymbol{\psi}^{\pi_\star}(s, a)^\top \boldsymbol{w}_{n+1}$, i.e.,

$$\boldsymbol{\theta}_{n+1}^{\mathrm{true}} = \boldsymbol{\theta}_{n+1} - \beta\gamma(y - \boldsymbol{\psi}^{\pi_\star}(s', a')^\top \boldsymbol{w}_{n+1}) \nabla_{\boldsymbol{\theta}_{n+1}} \mathbb{E}_{s'} \left[ \boldsymbol{\psi}^{\pi_{n+1}}(s', a'; \boldsymbol{\theta}_{n+1}) \right]^\top \boldsymbol{w}_{n+1}. \quad (5)$$

**Remark 1** *We remark on the difference in parameter updating between SFs and traditional RL. In SFs, the greedy target value with respect to $\boldsymbol{\theta}_{n+1}$ is guided by the current value ($\boldsymbol{\theta}_{n+1}^{\mathrm{new}}$ is guided by the parameter $\boldsymbol{\theta}_i$ v.s. $\boldsymbol{\theta}_{n+1}^{\mathrm{true}}$ is updated by the policy $\pi_\star$). In contrast, traditional RL leverages the target value to guide the current value. In conclusion, SFs updates the parameter within the target value, while traditional RL learns the parameter within the current value.*

In practical applications, SFs-estimation usually incorporates random errors $\varepsilon_Q$, which presumably stems from function approximation and is induced by source tasks. We rewrite

$$\boldsymbol{\psi}^{\pi_{n+1}}(s, a; \boldsymbol{\theta}_i)^\top \boldsymbol{w}_{n+1} = \boldsymbol{\psi}^{\pi_\star}(s, a; \boldsymbol{\theta}_\star)^\top \boldsymbol{w}_{n+1} + \varepsilon_Q. \tag{6}$$

Next, we state the underestimation bias of post-update SFs-estimate.

**Theorem 1** *Suppose that the agent has learn a policy set $\Pi = \{\pi_j\}_{j=1}^n$ of tasks $M_j \in \mathcal{M}^S$, $\mathcal{M}^S \subset \mathcal{M}^\phi$. For all $s \in \mathcal{S}$, $a \in \mathcal{A}$, let $\Psi = \{\boldsymbol{\psi}^{\pi_j}(s, a; \boldsymbol{\theta}_j)\}_{j=1}^n$ be the SFs set of $\Pi$ and $\pi_{n+1}$ be the DGPI policy defined in Eq. (2) on task $\boldsymbol{w}_{n+1}$. Assume $\boldsymbol{\psi}^{\pi_{n+1}}(s, a; \boldsymbol{\theta}_i)^\top \boldsymbol{w}_{n+1}$ satisfies Eq. (6), then the estimation bias satisfies*

$$\Delta(s', a') := \mathbb{E}_{\varepsilon_Q} \left[ \mathbb{E}_{s'} \left[ \boldsymbol{\psi}^{\pi_{n+1}}(s', a'; \boldsymbol{\theta}_{n+1}^{\text{new}}) \right]^\top \boldsymbol{w}_{n+1} - \mathbb{E}_{s'} \left[ \boldsymbol{\psi}^{\pi_{n+1}}(s', a'; \boldsymbol{\theta}_{n+1}^{\text{true}}) \right]^\top \boldsymbol{w}_{n+1} \right] \leq 0.$$

For detailed proof, please refer to Appendix C.1. Theorem 1 indicates that the estimation bias is labeled by underestimation, which extends the analysis of (Hunt et al., 2019, Theorem 3.1).

## 3.2 RELIEVING UNDERESTIMATION WITH DISTRIBUTIONAL SUCCESSOR FEATURES

Distributional RL based tool DSFs is utilized to relieve underestimation in this subsection. DSFs involves more information than the expectation of returns and subsequently learn Q-value accurately. First, we introduce some notations.

**Definition 1 (Distributional Successor Features)** *For a fixed features $\phi(s, a, s') \in \mathbb{R}^d$ and a policy $\pi$, the DSFs is a random vector that represents the sum of discounted features of the policy $\pi$, i.e., $D^\pi(s, a) := \sum_{i=t}^\infty \gamma^{i-t} \phi_{i+1}$. The DSFs can be computed through dynamic programming using a distributional Bellman operator,*

$$\mathcal{T}^\pi D(s, a) := \phi(s, a, s') + \gamma D(s', a'), \ s' \sim p(\cdot|s, a), a' \sim \pi(\cdot|s'). \tag{7}$$

Analogous to Eq. (1), for any weight vector $\boldsymbol{w}$, the distribution over returns is denoted by

$$Z(s, a) = D^\pi(s, a)^\top \boldsymbol{w}.$$

Note that in the approximation of $D^\pi(s, a)$, we use the same parameter symbols as $\psi^\pi(s, a)$ for convenience.

To establish the connection between Q-value and the distribution over returns, we incorporate different risk metrics into the decision-making. Analogous to the risk-sensitive RL framework (Ma et al., 2020; Zhou et al., 2023), we define a risk operator $\varphi : \mathcal{Z} \to \mathbb{R}$. The risk action-value is defined as $Q(s, a) = \varphi[Z(s, a)]$. Below, we list some common risk operators: the risk-neutral measure function $\varphi[\cdot] = \mathbb{E}[\cdot]$, mean-variance (Sobel, 1982; Tamar et al., 2012; Prashanth & Ghavamzadeh, 2016), value-at-risk (VaR) (Prashanth L & Fu, 2018), Wang's method (Wang, 2000), cumulative probability weighting parameterization (CPW) (Tversky & Kahneman, 1992) and conditional value at risk (CVaR) (Chow et al., 2015). Details are attached to Appendix F.3.

Based on DSFs, we can define DGPI by computing the risk action-value of transferring previous policies to a new task.

**Definition 2 (Distributional Generalized Policy Improvement)** *For all $(s, a) \in \mathcal{S} \times \mathcal{A}$, let $\mathcal{D} = \{D^{\pi_1}(s, a; \boldsymbol{\theta}_1), D^{\pi_2}(s, a; \boldsymbol{\theta}_2), \ldots, D^{\pi_n}(s, a; \boldsymbol{\theta}_n)\}$ be the DSFs of a set of policies. For a novel task, we can derive the DGPI policy $\pi_{n+1}(s) = \arg\max_a \max_{j \in [n]} \varphi \left[ D^{\pi_j}(s, a; \boldsymbol{\theta}_j)^\top \boldsymbol{w}_{n+1} \right]$.*

In contrast to the original GPI, DGPI provides more opportunities to leverage the information from source tasks adequately. To facilitate later analysis, we make the following assumption.

**Assumption 1** *Suppose that the random return $Z^{\pi_j}(s, a) = D^{\pi_j}(s, a)^\top \boldsymbol{w}_j$ can be approximated by the Gaussian distribution for all $\boldsymbol{w}_j \in \mathbb{R}^d, j \in [n]$, i.e., $Z^{\pi_j}(s, a) \sim \mathcal{N}(\varphi \left[ D^{\pi_j}(s, a; \boldsymbol{\theta}_j)^\top \boldsymbol{w}_j \right], \sigma^{\pi_j}(s, a; \boldsymbol{\xi}_j)^2)$ with parameters $\boldsymbol{\theta}_j$ and $\boldsymbol{\xi}_j$. Analogously, the random greedy target $f_j = \mathcal{T}^{\pi_j} D(s, a)^\top \boldsymbol{w}_j = \phi(s, a, s')^\top \boldsymbol{w}_j + \gamma D^{\pi_j}(s', a')^\top \boldsymbol{w}_j$, is approximated by $\mathcal{N}(y_D^j, \sigma^{\pi_j}(s, a; \boldsymbol{\xi}_j)^2)$, where $y_D^j = \mathbb{E}[f_j] = \mathbb{E} \left[ \phi(s, a, s')^\top \boldsymbol{w}_j \right] + \gamma \mathbb{E}_{s'}[\varphi[D^{\pi_j}(s', a'; \boldsymbol{\theta}_j)^\top \boldsymbol{w}_j]]$.*

In the distributional version, we use the same symbol in Eq. (6) to represent the random error

$$\varphi\left[D^{\pi_{n+1}}(s,a;\boldsymbol{\theta}_i)^\top \boldsymbol{w}_{n+1}\right] = \varphi\left[D^{\pi_\star}(s,a)^\top \boldsymbol{w}_{n+1}\right] + \epsilon_Q. \tag{8}$$

Now, we formally provide the estimation bias of post-update DSFs-estimate under the Gaussian assumption of value distribution.

**Theorem 2** *Suppose that Assumption 1 holds. For a new task $\boldsymbol{w}_{n+1}$, let*

$$\pi_{n+1}(s) = \arg\max_b \varphi\left[D^{\pi_i}(s,b;\boldsymbol{\theta}_i)^\top \boldsymbol{w}_{n+1}\right], \; i = \arg\max_{j\in[n]} \varphi\left[D^{\pi_j}(s,b;\boldsymbol{\theta}_j)^\top \boldsymbol{w}_{n+1}\right]$$

*be the DGPI policy obtained from DSFs set $\mathcal{D}$, and $\sigma^{\pi_{n+1}}(s,a;\boldsymbol{\xi}_i)^2$ be the variance of current value. Denote $\mathbb{E}_{s'}\left[\varphi\left[D^{\pi_{n+1}^{\text{new}}}(s',a';\boldsymbol{\theta}_{n+1}^{\text{new}})^\top \boldsymbol{w}_{n+1}\right]\right]$ and $\mathbb{E}_{s'}\left[\varphi\left[D^{\pi_{n+1}^{\text{true}}}(s',a';\boldsymbol{\theta}_{n+1}^{\text{true}})^\top \boldsymbol{w}_{n+1}\right]\right]$ by $\mathcal{Q}_{\boldsymbol{\theta}_{n+1}^{\text{new}}}$ and $\mathcal{Q}_{\boldsymbol{\theta}_{n+1}^{\text{true}}}$, respectively. Assume $\varphi\left[D^{\pi_{n+1}}(s,a;\boldsymbol{\theta}_i)^\top \boldsymbol{w}_{n+1}\right]$ satisfies Eq. (8), then the estimate bias of post-update $\mathcal{Q}_{\boldsymbol{\theta}_{n+1}^{\text{new}}}$ is*

$$\Delta_D(s',a') := \mathbb{E}_{\epsilon_Q}\left[\mathcal{Q}_{\boldsymbol{\theta}_{n+1}^{\text{new}}} - \mathcal{Q}_{\boldsymbol{\theta}_{n+1}^{\text{true}}}\right] = \frac{\Delta(s',a')}{\sigma^{\pi_{n+1}}(s,a;\boldsymbol{\xi}_i)^2},$$

*where $\boldsymbol{\theta}_{n+1}^{\text{true}}$ denotes the post-update parameters, which is obtained by the true current value $\varphi\left[D^{\pi_\star}(s,a)^\top \boldsymbol{w}_{n+1}\right]$.*

For detailed proof, please refer to Appendix C.2. Theorem 2 discloses the efficiency of DSFs:

- **Initial Training Stage:** Due to the disorder of exploration, $\varphi\left[D^{\pi_{n+1}}(s,a;\boldsymbol{\theta}_i)^\top \boldsymbol{w}_{n+1}\right]$ is lack of stability, which exhibits large fluctuation in $\left|\mathbb{E}_{\epsilon_Q}\left[\epsilon_Q\right]\right|$ and $\sigma^{\pi_{n+1}}(s,a;\boldsymbol{\xi}_i)^2$ (absolute mean and variance of $\epsilon_Q$). In this way, $\Delta_D(s',a')$ generally trends towards faster shrinkage than $\Delta(s',a')$. This stage implies the reduction of underestimation globally.

- **Converge Training Stage:** After a period of training, uniformly, $\left|\mathbb{E}_{\epsilon_Q}\left[\epsilon_Q\right]\right|$ is relatively small. A certain $\epsilon_Q$ that unexpectedly meets larger value, will be possessed with potentially large variance $\sigma^{\pi_{n+1}}(s,a;\boldsymbol{\xi}_i)^2$. In other words, DSFs exhibits higher efficiency in underestimation correction than SFs. This stage implies the reduction of underestimation individually.

Next, we will explore the generalization bound of DSFs without any distribution assumptions.

**Theorem 3** *(Generalization Bound of DSFs) Given a DSFs set $\mathcal{D}$, executing DGPI policy $\pi_{n+1}$ in task $\boldsymbol{w}_{n+1}$, we have*

$$\mathbb{E}\left[D^{\pi_\star}(s,a)^\top \boldsymbol{w}_{n+1}\right] - \varphi\left[D^{\pi_{n+1}}(s,a;\boldsymbol{\theta}_{n+1})^\top \boldsymbol{w}_{n+1}\right] \le \delta_\varphi + \frac{2}{1-\gamma}\phi_{\max}\min_{j\in[n]}\|\boldsymbol{w}_{n+1} - \boldsymbol{w}_j\|, \tag{9}$$

*where $\delta_\varphi = \mathbb{E}\left[D^{\pi_\star}(s,a)^\top \boldsymbol{w}_{n+1}\right] - \varphi\left[D^{\pi_\star}(s,a)^\top \boldsymbol{w}_{n+1}\right]$ and $\phi_{\max} = \max_{s,a}\|\boldsymbol{\phi}(s,a)\|$.*

**Proof Sketch.** By the triangle inequality, for any $j \in [n]$, we have

$$\mathbb{E}\left[D^{\pi_\star}(s,a)^\top \boldsymbol{w}_{n+1}\right] - \varphi\left[D^{\pi_{n+1}}(s,a;\boldsymbol{\theta}_{n+1})^\top \boldsymbol{w}_{n+1}\right]$$

$$= \underbrace{\mathbb{E}\left[D^{\pi_\star}(s,a)^\top \boldsymbol{w}_{n+1}\right] - \varphi\left[D^{\pi_\star}(s,a)^\top \boldsymbol{w}_{n+1}\right]}_{(A)}$$

$$+ \underbrace{\varphi\left[D^{\pi_\star}(s,a)^\top \boldsymbol{w}_{n+1}\right] - \varphi\left[D^{\pi_j}(s,a;\boldsymbol{\theta}_j)^\top \boldsymbol{w}_{n+1}\right]}_{(B)}$$

$$+ \underbrace{\varphi\left[D^{\pi_j}(s,a;\boldsymbol{\theta}_j)^\top \boldsymbol{w}_{n+1}\right] - \varphi\left[D^{\pi_{n+1}}(s,a;\boldsymbol{\theta}_{n+1})^\top \boldsymbol{w}_{n+1}\right]}_{(C)}.$$

In contrast to (Barreto et al., 2017), our result is established under the risk-sensitive setting and considers how the behavior of the risk operator affects the optimal policy in the new task, as characterized by (A). In other words, (A) represents the discrepancy of the random return between the

risk operator and the expectation operator. (B) and (C) can be dealt by Lemma 2 and Lemma 1, respectively. For detailed proof, please refer to Appendix C.3. □

Theorem 3 affords an alternative to mitigate underestimations (if $\delta_\varphi < 0$) caused by the limitation of the source tasks. The scenario of "mean-variance" ($\varphi[Z] = \mathbb{E}[Z] - \alpha\sigma[Z]$) is taken as an example, i.e., $\delta_\varphi = \alpha\sigma[D^{\pi_\star}(s,a)^\top \boldsymbol{w}_{n+1}]$. Obviously, $\delta_\varphi$ is positively correlated with parameter $\alpha$. Choosing $\alpha < 0$ leads to a lower generalization bound.

**Remark 2** *We remark that $\delta_\varphi > 0$ makes no focus. Essentially, the enlargement of an upper bound hardly provides any information on the choice of the risk operator, in contrast to reducing the upper bound. In other words, the decrease in the upper bound indicates that the error is limited to a smaller range, while the increase in the upper bound does not mean that the performance will deteriorate.*

Generally speaking, Theorem 2 verifies the superiority of DSFs in mitigating underestimation. Theorem 3 exhibits the potential of DSFs to reduce the generalization bound. In this way, a reasonable scheme introduces DSFs to mitigate underestimation and simultaneously possesses a lower generalization bound. The second term $2\boldsymbol{\phi}_{\max} \min_{j\in[n]} \|\boldsymbol{w}_{n+1} - \boldsymbol{w}_j\|/(1-\gamma)$ in Eq. (9) is focused. The agent will attain near-optimal performance, if the set of DSFs enough close to task $\boldsymbol{w}_{n+1}$. This presents a crucial question: How can we construct a set of DSFs to achieve optimal transfer for any new linearly-expressible task?

This question has been extensively explored in the MORL literature (Yang et al., 2019b; Hayes et al., 2022). According to Section 2.2, solving this question is equivalent to constructing a CCS by MORL methods. In the following, we will employ the DSF&DGPI technique to construct a CCS.

### 3.3 SUPERIOR CONVEX COVERAGE SET CONSTRUCTION FOR TRANSFER

Our goal is to construct a policy set $\Pi$ along with its corresponding DSFs set $\mathcal{D}$ for any tasks, by solving the following problem:

$$\arg\min_{\Pi} \mathbb{E}_{\boldsymbol{w}_{n+1}\sim \boldsymbol{W}^\phi} \left[\mathcal{L}(\pi_{n+1}, \boldsymbol{w}_{n+1})\right],$$

where $\mathcal{L}(\pi_{n+1}, \boldsymbol{w}_{n+1}) = \mathbb{E}\left[(D^{\pi_\star})^\top \boldsymbol{w}_{n+1}\right] - \varphi\left[(D^{\pi_{n+1}}_{\boldsymbol{\theta}_{n+1}})^\top \boldsymbol{w}_{n+1}\right]$ and $\boldsymbol{W}^\phi$ represents the weight set of $\mathcal{M}^\phi$. Recall that the expected SFs definition in Section 2.2, the expected DSFs can be defined as $D^\pi_{\boldsymbol{\theta}} := \mathbb{E}_{S_0\sim\mu}[D^\pi(S_0, \pi(S_0); \boldsymbol{\theta})]$ analogously. In practice, the expectation is over tasks $\boldsymbol{w}_{n+1}$ drawn uniformly at random from the set $\boldsymbol{W}^\phi$. Further, we can define the CCS based on the expected DSFs $D^\pi_{\boldsymbol{\theta}}$:

$$\text{CCS} = \{D^\pi_{\boldsymbol{\theta}} \mid \exists\boldsymbol{w} \text{ s.t. } \forall D^{\pi'}_{\boldsymbol{\theta}'}, \mathbb{E}[(D^\pi_{\boldsymbol{\theta}})^\top \boldsymbol{w}] \geq \mathbb{E}[(D^{\pi'}_{\boldsymbol{\theta}'})^\top \boldsymbol{w}]\}. \tag{10}$$

**Theorem 4** *Let $\Pi = \{\pi_j\}_{j=1}^n$ be a set of policies, and $\mathcal{D} = \{D^{\pi_j}_{\boldsymbol{\theta}_j}\}_{j=1}^n$ be the corresponding set of expected DSFs forming a CCS (Eq. (10)). Then, for any given weight vector $\boldsymbol{w}_{n+1}$, the DGPI policy $\pi_{n+1}$ is optimal with respect to $\boldsymbol{w}_{n+1}$: $\mathbb{E}\left[(D^{\pi_\star})^\top \boldsymbol{w}_{n+1}\right] = \varphi\left[(D^{\pi_{n+1}}_{\boldsymbol{\theta}_{n+1}})^\top \boldsymbol{w}_{n+1}\right].$*

For detailed proof, please refer to Appendix C.4. Theorem 4 indicates that the CCS constituted by the expected DSFs guarantees to access ability of the optimal policy using DGPI for any given task. However, the complete CCS is almost unachievable in real-world scenarios. At this moment, we consider constructing a weaker $\epsilon$-CCS, which can be induced by set max policy (SMP) $\pi^{\text{SMP}}_{n+1} = \arg\max_{j\in[n]} \varphi\left[(D^{\pi_j}_{\boldsymbol{\theta}_j})^\top \boldsymbol{w}_{n+1}\right]$ (Zahavy et al., 2021).

**Definition 3** *A DSFs set $\mathcal{D} = \{D^{\pi_j}_{\boldsymbol{\theta}_j}\}_{j=1}^n$ is an $\epsilon$-CCS if $\mathbb{E}[(D^{\pi_\star})^\top \boldsymbol{w}_{n+1}] - v^{\text{SMP}}_{\boldsymbol{w}_{n+1}} \leq \epsilon$ for all $\boldsymbol{w}_{n+1} \in \boldsymbol{W}^\phi$, where $v^{\text{SMP}}_{\boldsymbol{w}_{n+1}} = \max_{j\in[n]} \varphi\left[(D^{\pi_j}_{\boldsymbol{\theta}_j})^\top \boldsymbol{w}_{n+1}\right].$*

**Theorem 5** *Under the same assumptions as in Theorem 3. Given an $\epsilon_1$-CCS $\mathcal{D} = \{D^{\pi_j}_{\boldsymbol{\theta}_j}\}_{j=1}^n$ and a DGPI policy $\pi_{n+1}$ executed in task $\boldsymbol{w}_{n+1} \in \boldsymbol{W}^\phi$, then the obtained DSFs set satisfies $\epsilon_2$-CCS, where $\epsilon_2 \leq \min\left\{\epsilon_1, \delta_\varphi + \frac{2}{1-\gamma}\boldsymbol{\phi}_{\max} \max_{\boldsymbol{w}_{n+1}\in\boldsymbol{W}^\phi} \min_{j\in[n]} \|\boldsymbol{w}_{n+1} - \boldsymbol{w}_j\|\right\}.$*

For detailed proof, please refer to Appendix C.5. Theorem 5 discloses that DGPI narrows the incompleteness of the CCS set, and has the prospect of converging to CCS. In addition, similar to the analysis of Theorem 3, Theorem 5 implies that DGPI offers a greater potential to minimize the performance gap between the learned policy and the optimal policy for the new task, as compared to GPI (Alegre et al., 2022b).

## 4  IMPLEMENTATION DETAILS

In this section, we elaborate on how SFs-based algorithms incorporate DSFs. We first introduce the procedure of learning DSFs with DGPI, and finally provide its pseudocode. Details for RDSFOLS and DGPI-WCPI are deferred to Appendix D and Appendix E.

In Algorithm 1, we introduce an algorithm to learn DSFs with DGPI based on distributional value networks. Let $Z_\tau$ be the quantile function for the random variable $Z$, denoted by $Z_\tau = F_Z^{-1}(\tau) := \inf\{z \in \mathbb{R} : \tau \leq F_Z(z)\}$ (Müller, 1997). Note that $\tau$ represents the quantile fraction and it can be generated by different methods: quantile regression DQN (QR-DQN) (Dabney et al., 2018b), implicit quantile networks (IQN) (Dabney et al., 2018a) or fully parameterized quantile function (FQF) (Yang et al., 2019a). Generally, the training procedure of DSFDQN is identical in structure to SFDQN in (Barreto et al., 2017), except for the DQN update of successor features is replaced by the distributional RL algorithm. For two ensembles of quantile fractions $\{\hat{\tau}_j\}_{j=1}^N$ and $\{\hat{\tau}_k\}_{k=1}^N$, and policy $\pi$, the temporal difference (TD) error based on Eq. (7) is given by $\delta_t^{jk} = \phi_t^\top w + \gamma Z_{\hat{\tau}_j}^\pi(S_{t+1}, A_{t+1}; \boldsymbol{\theta}) - Z_{\hat{\tau}_k}^\pi(S_t, A_t; \boldsymbol{\theta}_i)$, where $Z^\pi(S_{t+1}, A_{t+1}; \boldsymbol{\theta}) = D^\pi(S_{t+1}, A_{t+1}; \boldsymbol{\theta})^\top w$ and $\hat{\tau} = (\tau_k + \tau_{k+1})/2, k = 1, \ldots, N$. We adopt quantile regression to train $Z_\tau^\pi(s, a; \boldsymbol{\theta})$ by minimizing the quantile Huber loss (Huber, 1964), with threshold $\kappa$,

$$\rho_\tau^\kappa(\delta_{jk}) = |\tau - \mathbb{I}\{\delta_{jk} < 0\}| \frac{\mathcal{L}_\kappa(\delta_{jk})}{\kappa}, \text{ with } \mathcal{L}_\kappa(\delta_{jk}) = \begin{cases} \frac{1}{2}\delta_{jk}^2, & \text{if } |\delta_{jk}| \leq \kappa, \\ \kappa\left(|\delta_{jk}| - \frac{1}{2}\kappa\right), & \text{otherwise.} \end{cases}$$

Then the objective of the quantile value network is defined as

$$J_Z(\boldsymbol{\theta}) = \sum_{j=0}^{N-1}\sum_{k=0}^{N-1}(\tau_{j+1} - \tau_j)\rho_{\hat{\tau}_k}^\kappa(\delta_{jk}^t).$$

According to Definition 2, a corresponding sample-based DGPI policy is obtained by $N$ samples of $\{\hat{\tau}_e\}_{e=1}^N$: $\pi(s) = \arg\max_a \max_i \varphi\left[Z_{\hat{\tau}_e}^{\pi_i}(s, a; \boldsymbol{\theta}_i)\right]$, where $Z_{\hat{\tau}_e}^{\pi_i}(s, a; \boldsymbol{\theta}_i) = D^{\pi_i}(s, a; \boldsymbol{\theta}_i)^\top w$.

---

**Algorithm 1** Learn DSFs with DGPI

1: **Initialize:** a set of policies $\Pi$ and the corresponding DSFs set $\mathcal{D}$, create $D^\pi$ parameterized by $\boldsymbol{\theta}$; replay buffer $\mathcal{B}$, weight $w$
2: Select initial state $s$ from $\mu$
3: **for** $t=0, \cdots$, num_step **do**
4:     Generate quantile fraction $\tau_e$
5:     **if** Bernoulli($\epsilon$) = 1 **then** $A_t \leftarrow$ Uniform($\mathcal{A}$)
6:     **else** $A_t \leftarrow \arg\max_b \max_i \varphi\left[D^{\pi_i}(S_t, b; \boldsymbol{\theta}_i)^\top w\right]$     // DGPI
7:     **end if**
8:     Execute $A_t$ and observe $\phi_t$ and $S_{t+1}$
9:     $\mathcal{B} \leftarrow \mathcal{B} \cup (S_t, A_t, \phi_t, S_{t+1})$
10:     **if** $S_{t+1}$ is not terminal **then** $S_t \leftarrow S_{t+1}$
11:     **else** select initial state $S_t$ from $\mu$
12:     **end if**
13:     Sample mini-batch $\{(s_k, a_k, \phi_k, s_k')\}_{k=1}^K$ from $\mathcal{B}$
14:     Generate quantile fractions $\tau_j, \tau_k$
15:     $a_k' \leftarrow \arg\max_b \varphi\left[D^\pi(s_k', b; \boldsymbol{\theta})^\top w\right]$ by $\tau_k$
16:     Update DSF by minimizing the loss $J_Z(\boldsymbol{\theta})$     // Learn DSF $D^\pi$
17: **end for**
18: **return** $\pi, D_{\boldsymbol{\theta}}^\pi$

---

## 5 Experiments

To evaluate the performance of DSFs, we conduct extensive quantitative evaluations in several environments, as shown in Fig. 1. First, Section 5.1 evaluates the efficiency of employing DSFs against SFs-based algorithms. A series of experiments under different risk scenarios are processed in Section 5.2. The detailed introduction for our experimental benchmark environments and additional experiment results are listed in Appendix F.

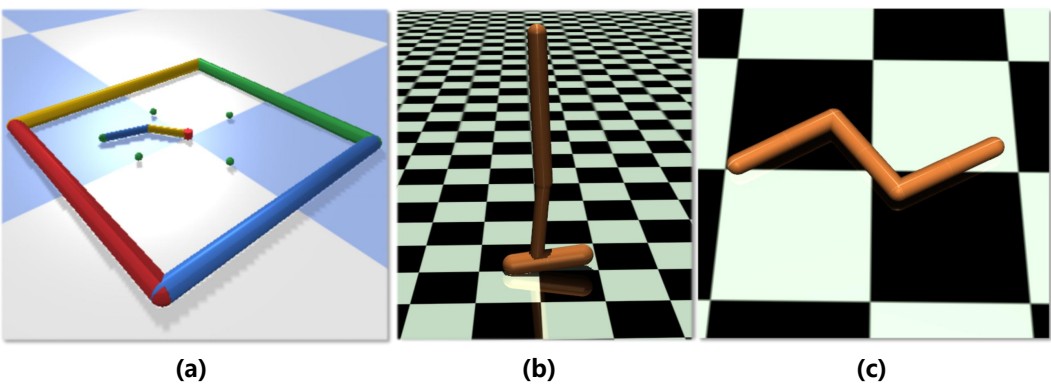

Figure 1: Example environments: (a) Reacher, (b) DiscreteHopper, (c) DiscreteSwimmer.

### 5.1 Evaluation on DSFs

First, we conduct a modular experiment to find the best way of quantile fraction generation. Compared with fix (QR-DQN) and net (FQF), random (IQN) has better performance and fewer parameters. Thus we use IQN in the following experiments. Due to space limitation, detailed comparison is postponed to Appendix F.2.

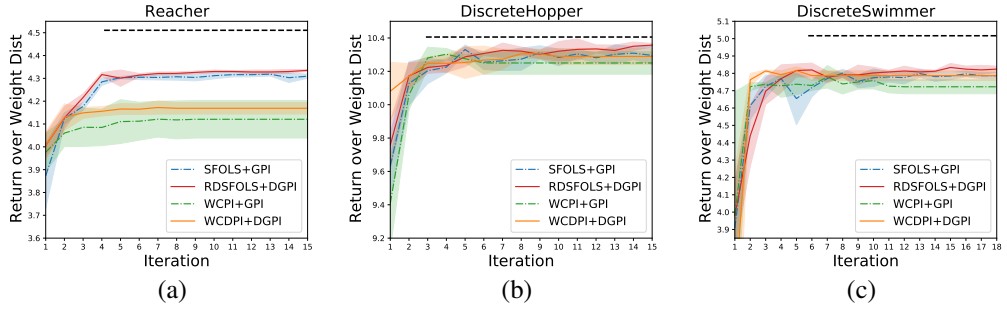

Figure 2: The expected return of the four algorithms over the task/reward weight distribution $W^\phi$. Each algorithm is run with five different random seeds. Each iteration is evaluated after 200000 steps per task. The black dashed line exhibits the true value of the last iteration.

We verify how the performance of DSFs outperforms SFs on the platform of MORL. SFOLS (Alegre et al., 2022b) and WCPI (Zahavy et al., 2021) are involved as two baselines for the comparison of our RDSFOLS and DGPI-WCPI.

In Fig. 2, we assess each method using a test set of 60 tasks uniformly sampled from $W^\phi$. The solid line represents the expected return (value) and the shaded region depicts the standard deviation. The results indicate that both the two DSFs-based algorithms (RDSFOLS and DGPI-WCPI) outperform SF-based methods (SFOLS and WCPI) in faster learning speed, higher return curve and lower variance. This is consistent with the comments on DSFs efficiency after Theorem 2.

We continue to observe a stably better performance of RDSFOLS against competing methods. This is presumably owing to the fact that RDSFOLS continuously improves its expected performances by

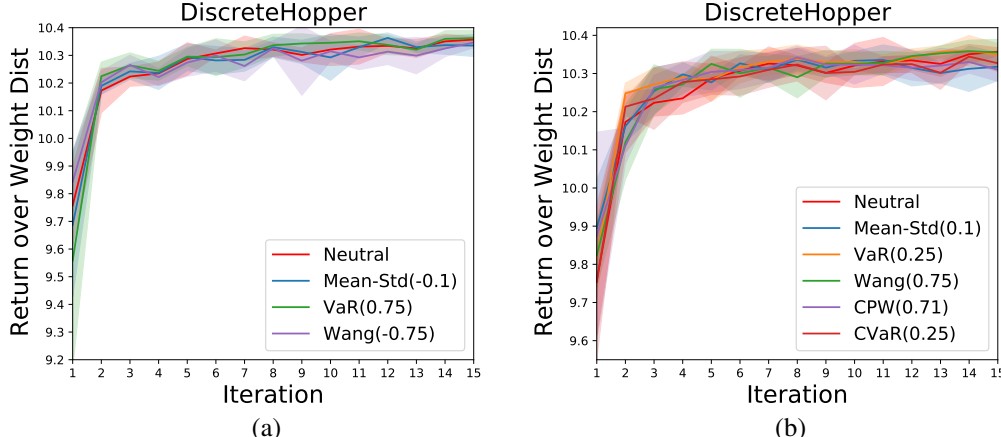

Figure 3: Comparison of the RDSFOLS algorithm using different risk operators within the Discrete-Hopper environment. Each curve is run with five seeds. (a) risk-seeking policies in DiscreteHopper, (b) risk-averse policies in DiscreteHopper.

learning new policies, rather than converging to a sub-optimal policy set like the WCPI algorithm; see the Reacher environment as an example.

## 5.2 POLICY TRAINING UNDER RISK OPERATORS

In this subsection, we investigate the influence of different risk operators on RDSFOLS. As Section 3.2 described, we conduct a comparative analysis involving three risk-seeking learned policies (mean-variance, Wang and VaR) in distributional RL with the risk-neutral measure function, as shown in Fig. 3(a). Additionally, we also evaluate five risk-averse learned policies: mean-variance, Wang, VaR, CVaR and CPW in distributional RL in Fig. 3(b). For a more detailed introduction of the risk metric and additional results under different risk operators, please refer to Appendix F.3 and Appendix F.4. Both risk-averse and risk-seeking methods exhibit little difference among all risk operators, which demonstrates the robustness of RDSFOLS.

## 6 CONCLUSION AND DISCUSSION

In this paper, we have undertaken a theoretical investigation into the underestimation in SF&GPI. Subsequently, we have introduced the concepts of DSFs and DGPI to tackle this challenge. The combination of distributional RL and SF&GPI relieves such underestimation and simultaneously narrows the generalization bound to some extent. Finally, we verify the quality of employing DSFs on the platform of MORL. As future work, we can integrate DSFs into universal successor features approximators (USFAs) (Borsa et al., 2018). This integration aims to enhance scalability, especially in high-dimensional problems.

Finally, related works on SFs, distributional RL and MORL are provided in Appendix A.

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

## APPENDIX

## A  RELATED WORK

### A.1  SUCCESSOR FEATURES

SF&GPI framework (Barreto et al., 2017; 2018), a popular approach to transfer in RL, aims to solve target tasks efficiently with minimal or even no additional learning by exploiting features

and policies from source tasks. Recent advancements in multi-tasking deep reinforcement learning agents leveraging successor features show promising results in enhancing sample efficiency (Kim et al., 2022; Machado et al., 2023).

Further, universal successor features approximators (USFAs) (Borsa et al., 2018) incorporate universal value function approximators (UVFAs) into the original successor features framework, allowing them to process policy vectors as inputs and thus enabling the use of GPI with arbitrary approximate policies. Then Nemecek & Parr (2021) established suboptimality bounds for assessing whether policies suffice the existing cache on a given performance threshold or if it justifies the need to learn a new policy. Meanwhile, Filos et al. (2021) introduced the inverse temporal difference learning (ITD) inverse RL algorithm, which simultaneously learns shared state features, alongside per-agent successor features and weight vectors. They integrated this approach with RL and GPI, leveraging non-rewarded demonstrations from other agents to expedite learning for the ego-agent. Recently, a modular neural network for learning features and SFs produced by their own modules called modular successor feature approximators (MSFA) (Carvalho et al., 2023) shows the significance of modularity in enabling reward-driven SF discovery. Transferring with SF&GPI guarantees that the resulting GPI policy performs at least as well as any of the source policies. However, GPI policy usually performs poorly in some situations. Hunt et al. (2019) observed that GPI consistently adopts a "pessimistic" approach, but they did not offer a comprehensive theoretical explanation. In this paper, we explicitly analyze the phenomenon of GPI underestimation, and introduce concepts of distributional RL and multi-objective RL to develop a novel approach that effectively tackles this problem.

## A.2 DISTRIBUTIONAL REINFORCEMENT LEARNING

Distributional RL has been extensively explored in research efforts to enhance performance, with a key challenge being the precise approximation of the value function distribution. Recent studies have made notable strides in addressing this challenge. Based on the framework of deep Q-network (DQN), Bellemare et al. (2017) introduced a technique for learning quantile values (or locations), either on a fixed uniform grid (Bellemare et al., 2017) or through sampled quantile fractions (Dabney et al., 2018b). Further, Dabney et al. (2018a) proposed a sampled quantile fractions method, called IQN. To avoid random samples, FQF (Yang et al., 2019a) parameterizes both the quantile values and quantile fractions, creating a fully parameterized tool for approximating distributions. Moreover, distributional RL provides a new perspective for optimizing policy by approximating the value function under diverse risk operators (Ma et al., 2020; Théate & Ernst, 2023). Duan et al. (2022) formally investigated the distributional return function in solving overestimation, which is the foundation of our work. Different from their analysis in a single task, we study the behavior of the distributional version in alleviating the underestimation problem within the SFs framework.

## A.3 MULTI-OBJECTIVE REINFORCEMENT LEARNING

Multi-objective reinforcement learning (MORL) tackles sequential decision problems involving agents with varying weights for potentially conflicting reward functions. Existing MORL algorithms can be broadly categorized into two primary groups: single-policy methods and multiple-policy methods (Van Moffaert et al., 2013).

Single-policy algorithms (Van Moffaert et al., 2013) seek to find the optimal policy by fixed weights induced scalarization of the multi-objective problem. While these methods provide the advantage of reduced computational cost, they usually require prior knowledge of objective weights, which can change over time.

Multiple-policy algorithms concentrate on learning a set of policies that approximate the true Pareto frontier. Deep optimistic linear support learning (DOL) (Mossalam et al., 2016) first applies deep RL algorithm to solve high-dimensional multi-objective decision problems. Yang et al. (2019b) introduced a multi-objective variant of the Bellman optimality operator, employing it to acquire a unified parametric representation of all optimal policies within the weight space, intending to enable few-shot adaptation of autonomous agents to new scenarios. Hayes et al. (2022) first discussed similarities between MORL and SFs. From the perspective of optimal updating, the authors claimed the superiority of MORL over SFs, owing to the fact that SFs constructs a scalar reward, and would lose partial information in real-world applications. Alegre et al. (2022b) integrated the MORL and SFs frameworks to introduce SFOLS, an extension of the optimistic linear support algorithm (Roijers

et al., 2015; Mossalam et al., 2016). SFOLS identifies tasks to solve, forming a convex coverage set (CCS) of SFs corresponding to their policies. Moreover, in cases where only partial CCS is accessible, they established an upper bound on the performance of the GPI policy. Differently, our paper provides the improvement upon a set of policies by DGPI to narrow the incompleteness of the CCS set, and subsequently minimizes the performance gap between the DGPI policy and the optimal policy for the new task.

## B    DISTRIBUTIONAL REINFORCEMENT LEARNING

Instead of a scalar value function $Q^\pi(s, a)$, we estimate the distribution over returns (the law of random variable return $Z^\pi(s, a) = \sum_{t=0}^{\infty} \gamma^t R(S_t, A_t)$) in distributional RL. Note that $Q^\pi(s, a) = \mathbb{E}[Z^\pi(s, a)]$. As with the standard RL, the distributional Bellman operator for policy evaluation is

$$\mathcal{T}^\pi Z(s, a) \stackrel{D}{=} R(s, a) + \gamma Z(s', a'), \ s' \sim p(\cdot | s, a), a' \sim \pi(\cdot | s'),$$

where $U :\stackrel{D}{=} V$ denotes that random variables $U$ and $V$ have equal probability laws. $\mathcal{T}^\pi$ is proved to be a contraction in the $p$-Wasserstein distance (Bellemare et al., 2017).

## C    PROOFS

For simplicity, we denote $\boldsymbol{\psi}(s, a)^\top = \varphi\left[D(s, a)^\top \boldsymbol{w}\right] \boldsymbol{w}^\top (\boldsymbol{w}\boldsymbol{w}^\top)^{-1}$, where $\boldsymbol{w} \in \mathbb{R}^d$.

### C.1    PROOF OF THEOREM 1

*Proof.* Supposing that $\beta$ is sufficiently small, we can approximate the post-update Q-function effectively according to the post-update parameters $\boldsymbol{\theta}_{n+1}^{\text{new}}, \boldsymbol{\theta}_{n+1}^{\text{true}}$ equations (as expressed in Eq. (4) and Eq. (5)) and Taylor's expansion.

$$\mathbb{E}_{s'}\left[\boldsymbol{\psi}^{\pi_{n+1}}(s', a'; \boldsymbol{\theta}_{n+1}^{\text{new}})\right]^\top \boldsymbol{w}_{n+1}$$
$$\approx \mathbb{E}_{s'}\left[\boldsymbol{\psi}^{\pi_{n+1}}(s', a'; \boldsymbol{\theta}_{n+1})\right]^\top \boldsymbol{w}_{n+1}$$
$$- \beta\gamma(y - \boldsymbol{\psi}^{\pi_{n+1}}(s', a'; \boldsymbol{\theta}_i)^\top \boldsymbol{w}_{n+1}) \|\nabla_{\boldsymbol{\theta}_{n+1}} \mathbb{E}_{s'}\left[\boldsymbol{\psi}^{\pi_{n+1}}(s', a'; \boldsymbol{\theta}_{n+1})\right]^\top \boldsymbol{w}_{n+1}\|_2^2,$$

$$\mathbb{E}_{s'}\left[\boldsymbol{\psi}^{\pi_{n+1}}(s', a'; \boldsymbol{\theta}_{n+1}^{\text{true}})\right]^\top \boldsymbol{w}_{n+1}$$
$$\approx \mathbb{E}_{s'}\left[\boldsymbol{\psi}^{\pi_{n+1}}(s', a'; \boldsymbol{\theta}_{n+1})\right]^\top \boldsymbol{w}_{n+1}$$
$$- \beta\gamma(y - \boldsymbol{\psi}^{\pi_\star}(s', a')^\top \boldsymbol{w}_{n+1}) \|\nabla_{\boldsymbol{\theta}_{n+1}} \mathbb{E}_{s'}\left[\boldsymbol{\psi}^{\pi_{n+1}}(s', a'; \boldsymbol{\theta}_{n+1})\right]^\top \boldsymbol{w}_{n+1}\|_2^2.$$

Notice that $\boldsymbol{\psi}^{\pi_{n+1}}(s, a; \boldsymbol{\theta}_i)^\top \boldsymbol{w}_{n+1} = \boldsymbol{\psi}^{\pi_\star}(s, a)^\top \boldsymbol{w}_{n+1} + \varepsilon_Q$. Then, in expectation over $\varepsilon_Q$, the estimation bias of post-update Q-estimate $\mathbb{E}_{s'}\left[\boldsymbol{\psi}^{\pi_{n+1}}(s', a'; \boldsymbol{\theta}_{n+1}^{\text{new}})\right]^\top \boldsymbol{w}_{n+1}$ is

$$\Delta(s', a') = \mathbb{E}_{\varepsilon_Q}\left[\mathbb{E}_{s'}\left[\boldsymbol{\psi}^{\pi_{n+1}}(s', a'; \boldsymbol{\theta}_{n+1}^{\text{new}})\right]^\top \boldsymbol{w}_{n+1} - \mathbb{E}_{s'}\left[\boldsymbol{\psi}^{\pi_{n+1}}(s', a'; \boldsymbol{\theta}_{n+1}^{\text{true}})\right]^\top \boldsymbol{w}_{n+1}\right]$$
$$\approx \beta\gamma\mathbb{E}_{\varepsilon_Q}\left[\boldsymbol{\psi}^{\pi_{n+1}}(s', a'; \boldsymbol{\theta}_i)^\top \boldsymbol{w}_{n+1} - \boldsymbol{\psi}^{\pi_\star}(s', a')^\top \boldsymbol{w}_{n+1}\right]$$
$$\cdot \|\nabla_{\boldsymbol{\theta}_{n+1}} \mathbb{E}_{s'}\left[\boldsymbol{\psi}^{\pi_{n+1}}(s', a'; \boldsymbol{\theta}_{n+1})\right]^\top \boldsymbol{w}_{n+1}\|_2^2$$
$$= \beta\gamma\mathbb{E}_{\varepsilon_Q}\left[\varepsilon_Q\right] \cdot \|\nabla_{\boldsymbol{\theta}_{n+1}} \mathbb{E}_{s'}\left[\boldsymbol{\psi}^{\pi_{n+1}}(s', a'; \boldsymbol{\theta}_{n+1})\right]^\top \boldsymbol{w}_{n+1}\|_2^2.$$

Previous research (Li et al., 2022; Hunt et al., 2019) showed that the DGPI policy $\pi_{n+1}$ always acts according to a lower bound (Eq. (3)) on the action-value. In other words, $\mathbb{E}_{\varepsilon_Q}\left[\varepsilon_Q\right] \leq 0$. Therefore, it is clear that $\Delta(s', a') \leq 0$. $\quad\square$

### C.2    PROOF OF THEOREM 2

*Proof.* For a constant weight $\boldsymbol{w}_{n+1}$, we first define a greedy target $y_D^{n+1} = \mathbb{E}[\boldsymbol{\phi}(s, a)^\top \boldsymbol{w}_{n+1}] + \gamma\mathbb{E}_{s'}\left[\varphi\left[D^{\pi_{n+1}}(s', a'; \boldsymbol{\theta}_{n+1})^\top \boldsymbol{w}_{n+1}\right]\right]$, where $a' = \arg\max_b \boldsymbol{\psi}^{\pi_{n+1}}(s, b; \boldsymbol{\theta}_{n+1})^\top \boldsymbol{w}_{n+1}$. Suppose that $y_D^{n+1}$ obeys a Gaussian distribution $\mathcal{Z}^{\text{target}}(\cdot | s, a)$. Since $\mathbb{E}[y_D^{n+1}]$ is equal to $y$ in Eq.

(4), it follows that $\mathcal{Z}^{\text{target}}(\cdot|s,a) = \mathcal{N}(y_D^{n+1}, \sigma^{\pi_{n+1}}(s,a;\boldsymbol{\xi}_{n+1})^2)$. Let the current value obeys the Gaussian distribution $\mathcal{Z}(\cdot|s,a) = \mathcal{N}(\boldsymbol{\psi}^{\pi_{n+1}}(s,a;\boldsymbol{\theta}_i)^\top \boldsymbol{w}_{n+1}, \sigma^{\pi_{n+1}}(s,a;\boldsymbol{\xi}_i)^2)$. Then the DSFs-estimate $D^{\pi_{n+1}}(s,a;\boldsymbol{\theta}_{n+1})$ can be updated by minimizing the Kullback–Leibler (KL) divergence $D_{KL}(\mathcal{Z}^{\text{target}}(\cdot|s,a), \mathcal{Z}(\cdot|s,a))$:

$$\boldsymbol{\theta}_{n+1}^{\text{new}} = \boldsymbol{\theta}_{n+1} - \beta \frac{y_D^{n+1} - \boldsymbol{\psi}^{\pi_{n+1}}(s,a;\boldsymbol{\theta}_i)^\top \boldsymbol{w}_{n+1}}{\sigma^{\pi_{n+1}}(s,a;\boldsymbol{\xi}_i)^2} \gamma \nabla_{\boldsymbol{\theta}_{n+1}} \mathbb{E}_{s'} \left[ \varphi \left[ D^{\pi_{n+1}}(s',a';\boldsymbol{\theta}_{n+1})^\top \boldsymbol{w}_{n+1} \right] \right].$$

Similarly, for $\boldsymbol{\theta}_{n+1}^{\text{true}}$, we have

$$\boldsymbol{\theta}_{n+1}^{\text{new}} = \boldsymbol{\theta}_{n+1} - \beta \frac{y_D^{n+1} - \boldsymbol{\psi}^{\pi_{n+1}}(s,a)^\top \boldsymbol{w}_{n+1}}{\sigma^{\pi_{n+1}}(s,a;\boldsymbol{\xi}_i)^2} \gamma \nabla_{\boldsymbol{\theta}_{n+1}} \mathbb{E}_{s'} \left[ \varphi \left[ D^{\pi_{n+1}}(s',a';\boldsymbol{\theta}_{n+1})^\top \boldsymbol{w}_{n+1} \right] \right].$$

Again by linearizing around $\boldsymbol{\theta}_{n+1}$ using Taylor's expansion, we obtain

$$\mathbb{E}_{s'} \left[ \boldsymbol{\psi}^{\pi_{n+1}}(s',a';\boldsymbol{\theta}_{n+1}^{\text{new}}) \right]^\top \boldsymbol{w}_{n+1}$$
$$\approx \mathbb{E}_{s'} \left[ \boldsymbol{\psi}^{\pi_{n+1}}(s',a';\boldsymbol{\theta}_{n+1}) \right]^\top \boldsymbol{w}_{n+1}$$
$$- \beta\gamma \frac{y_D^{n+1} - \boldsymbol{\psi}^{\pi_{n+1}}(s,a;\boldsymbol{\theta}_i)^\top \boldsymbol{w}_{n+1}}{\sigma^{\pi_{n+1}}(s,a;\boldsymbol{\xi}_i)^2} \| \nabla_{\boldsymbol{\theta}_{n+1}} \mathbb{E}_{s'} \left[ \varphi \left[ D^{\pi_{n+1}}(s',a';\boldsymbol{\theta}_{n+1})^\top \boldsymbol{w}_{n+1} \right] \right] \|_2^2,$$

$$\mathbb{E}_{s'} \left[ \boldsymbol{\psi}^{\pi_{n+1}}(s',a';\boldsymbol{\theta}_{n+1}^{\text{true}}) \right]^\top \boldsymbol{w}_{n+1}$$
$$\approx \mathbb{E}_{s'} \left[ \boldsymbol{\psi}^{\pi_{n+1}}(s',a';\boldsymbol{\theta}_{n+1}) \right]^\top \boldsymbol{w}_{n+1}$$
$$- \beta\gamma \frac{y_D^{n+1} - \boldsymbol{\psi}^{\pi_{n+1}}(s,a)^\top \boldsymbol{w}_{n+1}}{\sigma^{\pi_{n+1}}(s,a;\boldsymbol{\xi}_i)^2} \| \nabla_{\boldsymbol{\theta}_{n+1}} \mathbb{E}_{s'} \left[ \varphi \left[ D^{\pi_{n+1}}(s',a';\boldsymbol{\theta}_{n+1})^\top \boldsymbol{w}_{n+1} \right] \right] \|_2^2.$$

Therefore,

$$\Delta_D(s',a') = \mathbb{E}_{\epsilon_Q} \left[ \mathbb{E}_{s'} \left[ \boldsymbol{\psi}^{\pi_{n+1}}(s',a';\boldsymbol{\theta}_{n+1}^{\text{new}}) \right]^\top \boldsymbol{w}_{n+1} - \mathbb{E}_{s'} \left[ \boldsymbol{\psi}^{\pi_{n+1}}(s',a';\boldsymbol{\theta}_{n+1}^{\text{true}}) \right]^\top \boldsymbol{w}_{n+1} \right]$$

$$\approx \beta\gamma \mathbb{E}_{\epsilon_Q} \left[ \frac{\boldsymbol{\psi}^{\pi_{n+1}}(s',a';\boldsymbol{\theta}_i)^\top \boldsymbol{w}_{n+1}}{\sigma^{\pi_{n+1}}(s,a;\boldsymbol{\xi}_i)^2} - \frac{\boldsymbol{\psi}^{\pi_\star}(s',a')^\top \boldsymbol{w}_{n+1}}{\sigma^{\pi_{n+1}}(s,a;\boldsymbol{\xi}_i)^2} \right]$$
$$\cdot \| \mathbb{E}_{s'} \left[ \varphi \left[ D^{\pi_{n+1}}(s',a';\boldsymbol{\theta}_{n+1})^\top \boldsymbol{w}_{n+1} \right] \right] \|_2^2$$
$$= \frac{\beta\gamma \mathbb{E}_{\epsilon_Q} \left[ \epsilon_Q \right] \| \nabla_{\boldsymbol{\theta}_{n+1}} \mathbb{E}_{s'} \left[ \boldsymbol{\psi}^{\pi_{n+1}}(s',a';\boldsymbol{\theta}_{n+1}) \right]^\top \boldsymbol{w}_{n+1} \|_2^2}{\sigma^{\pi_{n+1}}(s,a;\boldsymbol{\xi}_i)^2}$$
$$= \frac{\Delta(s',a')}{\sigma^{\pi_{n+1}}(s,a;\boldsymbol{\xi}_i)^2}.$$

Then we complete the proof. $\square$

## C.3 PROOF OF THEOREM 3

### C.3.1 SOME IMPORTANT LEMMAS FOR PROVING THEOREM 3

**Lemma 1** *Let $\pi_1,\ldots,\pi_n$ be $n$ policies with risk action-value functions $\varphi[D^{\pi_j}(s,a;\boldsymbol{\theta}_j)^\top \boldsymbol{w}_{n+1}]$, $j \in [n]$. For a novel task $\boldsymbol{w}_{n+1}$, executing DGPI policy $\pi_{n+1}$, then we have*
$$\varphi[D^{\pi_{n+1}}(s,a;\boldsymbol{\theta}_{n+1})^\top \boldsymbol{w}_{n+1}] \geq \max_{j\in[n]} \varphi[D^{\pi_j}(s,a;\boldsymbol{\theta}_j)^\top \boldsymbol{w}_{n+1}].$$

*Proof.* For brevity we denote $\varphi_{\max}(s,a) = \max_{j\in[n]} \varphi[D^{\pi_j}(s,a;\boldsymbol{\theta}_j)^\top \boldsymbol{w}_{n+1}]$. For all $s \in \mathcal{S}, a \in \mathcal{A}$ and $j \in [n]$, we have

$$\mathcal{T}^{\pi_{n+1}} \varphi_{\max}(s,a) = \mathbb{E} \left[ \boldsymbol{\phi}(s,a,s')^\top \boldsymbol{w}_{n+1} \right] + \gamma \mathbb{E}_{s'} \left[ \varphi_{\max}(s',\pi_{n+1}(s')) \right]$$

$$= r_{\boldsymbol{w}_{n+1}}(s,a,s') + \gamma \mathbb{E}_{s'} \left[ \max_b \varphi_{\max}(s',b) \right]$$

$$\geq r_{\boldsymbol{w}_{n+1}}(s,a,s') + \gamma \mathbb{E}_{s'} \left[ \varphi_{\max}(s',\pi_j(s')) \right]$$

$$\geq r_{\boldsymbol{w}_{n+1}}(s,a,s') + \gamma \mathbb{E}_{s'} \left[ \varphi \left[ D^{\pi_j}(s',a';\boldsymbol{\theta}_j)^\top \boldsymbol{w}_{n+1} \right] \right]$$

$$= \mathcal{T}^{\pi_j} \varphi \left[ D^{\pi_j}(s,a;\boldsymbol{\theta}_j)^\top \boldsymbol{w}_{n+1} \right]$$

$$= \varphi \left[ D^{\pi_j}(s,a;\boldsymbol{\theta}_j)^\top \boldsymbol{w}_{n+1} \right].$$

Using the contraction and monotonicity of the Bellman operator $\mathcal{T}^{\pi_{n+1}}$ we have

$$\varphi\left[D^{\pi_{n+1}}(s,a;\boldsymbol{\theta}_{n+1})^{\top}\boldsymbol{w}_{n+1}\right] = \lim_{k\to\infty}(\mathcal{T}^{\pi_{n+1}})^{k}\varphi_{\max}(s,a)$$
$$\geq \varphi\left[D^{\pi_j}(s,a;\boldsymbol{\theta}_j)^{\top}\boldsymbol{w}_{n+1}\right].$$

$\square$

**Lemma 2** *Let $\phi_{\max} = \max_{s,a}\|\phi(s,a)\|_2$. Then*

$$\varphi[D^{\pi_\star}(s,a)^{\top}\boldsymbol{w}_{n+1}] - \varphi[D^{\pi_j}(s,a;\boldsymbol{\theta}_j)^{\top}\boldsymbol{w}_{n+1}] \leq \frac{2}{1-\gamma}\phi_{\max}\min_{j\in[n]}\|\boldsymbol{w}_{n+1} - \boldsymbol{w}_j\|.$$

*Proof.* By the triangle inequality, we get

$$\varphi[D^{\pi_{n+1}}(s,a;\boldsymbol{\theta}_{n+1})^{\top}\boldsymbol{w}_{n+1}] - \varphi[D^{\pi_j}(s,a;\boldsymbol{\theta}_j)^{\top}\boldsymbol{w}_{n+1}]$$
$$\leq \underbrace{\left|\varphi[D^{\pi_{n+1}}(s,a;\boldsymbol{\theta}_{n+1})^{\top}\boldsymbol{w}_{n+1}] - \varphi[D^{\pi_j}(s,a;\boldsymbol{\theta}_j)^{\top}\boldsymbol{w}_j]\right|}_{(A)}$$
$$+ \underbrace{\left|\varphi[D^{\pi_j}(s,a;\boldsymbol{\theta}_j)^{\top}\boldsymbol{w}_j] - \varphi[D^{\pi_j}(s,a;\boldsymbol{\theta}_j)^{\top}\boldsymbol{w}_{n+1}]\right|}_{(B)}. \tag{11}$$

For $j\in[n]$, let

$$e_j = \max_{s,a}\left|\varphi[D^{\pi_{n+1}}(s,a;\boldsymbol{\theta}_{n+1})^{\top}\boldsymbol{w}_{n+1}] - \varphi[D^{\pi_j}(s,a;\boldsymbol{\theta}_j)^{\top}\boldsymbol{w}_j]\right|.$$

Therefore, term (A) in Eq. (11) can be processed by

$$\left|\varphi[D^{\pi_{n+1}}(s,a;\boldsymbol{\theta}_{n+1})^{\top}\boldsymbol{w}_{n+1}] - \varphi[D^{\pi_j}(s,a;\boldsymbol{\theta}_j)^{\top}\boldsymbol{w}_j]\right|$$
$$= \left|\boldsymbol{\phi}(s,a,s')^{\top}\boldsymbol{w}_{n+1} + \gamma\mathbb{E}_{s'}\left[\max_b \varphi[D^{\pi_{n+1}}(s,b;\boldsymbol{\theta}_{n+1})^{\top}\boldsymbol{w}_{n+1}]\right.\right.$$
$$\left.\left. - \boldsymbol{\phi}(s,a,s')^{\top}\boldsymbol{w}_j - \gamma\mathbb{E}_{s'}\left[\max_b \varphi[D^{\pi_j}(s,b;\boldsymbol{\theta}_j)^{\top}\boldsymbol{w}_j]\right]\right|\right.$$
$$= \left|\boldsymbol{\phi}(s,a,s')^{\top}\boldsymbol{w}_{n+1} - \boldsymbol{\phi}(s,a,s')^{\top}\boldsymbol{w}_j\right.$$
$$\left. + \gamma\mathbb{E}_{s'}\left[\max_b \varphi[D^{\pi_{n+1}}(s,b;\boldsymbol{\theta}_{n+1})^{\top}\boldsymbol{w}_{n+1}] - \max_b \varphi[D^{\pi_j}(s,b;\boldsymbol{\theta}_j)^{\top}\boldsymbol{w}_j]\right]\right|$$
$$\leq \left|\boldsymbol{\phi}(s,a,s')^{\top}\boldsymbol{w}_{n+1} - \boldsymbol{\phi}(s,a,s')^{\top}\boldsymbol{w}_j\right|$$
$$+ \gamma\sum_{s'}p(s'|s,a)\left|\max_b \varphi[D^{\pi_{n+1}}(s,b;\boldsymbol{\theta}_{n+1})^{\top}\boldsymbol{w}_{n+1}] - \max_b \varphi[D^{\pi_j}(s,b;\boldsymbol{\theta}_j)^{\top}\boldsymbol{w}_j]\right|$$
$$\leq \phi_{\max}|\boldsymbol{w}_{n+1} - \boldsymbol{w}_j| + \gamma e_j. \tag{12}$$

Since Eq. (12) is valid for any $s,a\in\mathcal{S}\times\mathcal{A}$, we have shown that $e_j \leq \phi_{\max}|\boldsymbol{w}_{n+1} - \boldsymbol{w}_j| + \gamma e_j$. Solving for $e_j$ we obtain

$$e_j \leq \frac{1}{1-\gamma}\phi_{\max}|\boldsymbol{w}_{n+1} - \boldsymbol{w}_j|. \tag{13}$$

We now investigate the bound of (B) in Eq. (11). Let

$$e'_j = \max_{s,a}\left|\varphi[D^{\pi_j}(s,a;\boldsymbol{\theta}_j)^{\top}\boldsymbol{w}_j] - \varphi[D^{\pi_j}(s,a;\boldsymbol{\theta}_j)^{\top}\boldsymbol{w}_{n+1}]\right|.$$

Then we have

$$
\left| \varphi[D^{\pi_j}(s,a;\boldsymbol{\theta}_j)^\top \boldsymbol{w}_j] - \varphi[D^{\pi_j}(s,a;\boldsymbol{\theta}_j)^\top \boldsymbol{w}_{n+1}] \right|
$$

$$
= \left| \boldsymbol{\phi}(s,a,s')^\top \boldsymbol{w}_j + \gamma \mathbb{E}_{s'} \left[ \max_b \varphi[D^{\pi_j}(s,b;\boldsymbol{\theta}_j)^\top \boldsymbol{w}_j] \right. \right.
$$

$$
\left. \left. - \boldsymbol{\phi}(s,a,s')^\top \boldsymbol{w}_{n+1} - \gamma \mathbb{E}_{s'} \left[ \max_b \varphi[D^{\pi_j}(s,b;\boldsymbol{\theta}_j)^\top \boldsymbol{w}_{n+1}] \right] \right| \right.
$$

$$
= \left| \boldsymbol{\phi}(s,a,s')^\top \boldsymbol{w}_j - \boldsymbol{\phi}(s,a,s')^\top \boldsymbol{w}_{n+1} \right.
$$

$$
\left. + \gamma \mathbb{E}_{s'} \left[ \max_b \varphi[D^{\pi_j}(s,b;\boldsymbol{\theta}_j)^\top \boldsymbol{w}_j] - \max_b \varphi[D^{\pi_j}(s,b;\boldsymbol{\theta}_j)^\top \boldsymbol{w}_{n+1}] \right] \right|
$$

$$
\leq \left| \boldsymbol{\phi}(s,a,s')^\top \boldsymbol{w}_j - \boldsymbol{\phi}(s,a,s')^\top \boldsymbol{w}_{n+1} \right|
$$

$$
+ \gamma \sum_{s'} p(s'|s,a) \left| \max_b \varphi[D^{\pi_j}(s,b;\boldsymbol{\theta}_j)^\top \boldsymbol{w}_j] - \max_b \varphi[D^{\pi_j}(s,b;\boldsymbol{\theta}_j)^\top \boldsymbol{w}_{n+1}] \right|
$$

$$
\leq \phi_{\max} |\boldsymbol{w}_{n+1} - \boldsymbol{w}_j| + \gamma e'_j.
$$

Solving $e'_j$ as above, we derive that

$$
e'_j \leq \frac{1}{1-\gamma} \phi_{\max} |\boldsymbol{w}_{n+1} - \boldsymbol{w}_j|. \tag{14}
$$

Plugging Eq. (13) and Eq. (14) into Eq. (11), we complete the proof. □

### C.3.2 PROOF OF THEOREM 3

*Proof.* We follow the proof of (Barreto et al., 2017, Theorem 2), with additional error terms to account for the performance difference resulting from the risk action-value as opposed to the expectation value.

$$
\mathbb{E}\left[ D^{\pi_\star}(s,a)^\top \boldsymbol{w}_{n+1} \right] - \varphi\left[ D^{\pi_{n+1}}(s,a;\boldsymbol{\theta}_{n+1})^\top \boldsymbol{w}_{n+1} \right]
$$

$$
\leq \mathbb{E}\left[ D^{\pi_\star}(s,a)^\top \boldsymbol{w}_{n+1} \right] - \varphi\left[ D^{\pi_\star}(s,a)^\top \boldsymbol{w}_{n+1} \right]
$$

$$
+ \varphi\left[ D^{\pi_\star}(s,a)^\top \boldsymbol{w}_{n+1} \right] - \varphi\left[ D^{\pi_j}(s,a;\boldsymbol{\theta}_j)^\top \boldsymbol{w}_{n+1} \right] \qquad \text{(Lemma 2)}
$$

$$
+ \varphi\left[ D^{\pi_j}(s,a;\boldsymbol{\theta}_j)^\top \boldsymbol{w}_{n+1} \right] - \varphi\left[ D^{\pi_{n+1}}(s,a;\boldsymbol{\theta}_{n+1})^\top \boldsymbol{w}_{n+1} \right] \qquad \text{(Lemma 1)}
$$

$$
\leq \delta_\varphi + \frac{2}{1-\gamma} \phi_{\max} \min_{j \in [n]} \|\boldsymbol{w}_{n+1} - \boldsymbol{w}_j\|,
$$

where $\delta_\varphi = \mathbb{E}\left[ D^{\pi_\star}(s,a)^\top \boldsymbol{w}_{n+1} \right] - \varphi\left[ D^{\pi_\star}(s,a)^\top \boldsymbol{w}_{n+1} \right]$. □

## C.4 PROOF OF THEOREM 4

*Proof.* For all $s$, we have

$$
\varphi[D^{\pi_{n+1}}_{\boldsymbol{\theta}_{n+1}}(s)^\top \boldsymbol{w}_{n+1}] = \varphi[D^{\pi_{n+1}}(s,\pi_{n+1}(s);\boldsymbol{\theta}_{n+1})^\top \boldsymbol{w}_{n+1}]
$$

$$
\geq \max_{j \in [n], a \in \mathcal{A}} \varphi[D^{\pi_j}(s,a;\boldsymbol{\theta}_j)^\top \boldsymbol{w}_{n+1}]
$$

$$
\geq \max_{j \in [n]} \varphi[D^{\pi_j}_{\boldsymbol{\theta}_j}(s)^\top \boldsymbol{w}_{n+1}].
$$

Taking the expected value with respect to the initial state distribution $\mu$ on both sides, we obtain:

$$
\mathbb{E}_{S_0 \sim \mu} \left[ \varphi[D^{\pi_{n+1}}_{\boldsymbol{\theta}_{n+1}}(S_0)^\top \boldsymbol{w}_{n+1}] \right] \geq \mathbb{E}_{S_0 \sim \mu} \left[ \max_{j \in [n]} \varphi[D^{\pi_j}_{\boldsymbol{\theta}_j}(S_0)^\top \boldsymbol{w}_{n+1}] \right]
$$

$$
\geq \max_{j \in [n]} \mathbb{E}_{S_0 \sim \mu} \left[ \varphi[D^{\pi_j}_{\boldsymbol{\theta}_j}(S_0)^\top \boldsymbol{w}_{n+1}] \right]
$$

Recall that $\Pi$ is a set of policies and $\mathcal{D}$ is the corresponding set of expected DSFs, which is a CCS. In this case, $v_{\boldsymbol{w}_{n+1}}^{\mathrm{SMP}} = \mathbb{E}[(D^{\pi_\star})^\top \boldsymbol{w}_{n+1}]$. Then we have

$$\varphi[(D_{\boldsymbol{\theta}_{n+1}}^{\pi_{n+1}})^\top \boldsymbol{w}_{n+1}] \geq v_{\boldsymbol{w}_{n+1}}^{\mathrm{SMP}} = \mathbb{E}[(D^{\pi_\star})^\top \boldsymbol{w}_{n+1}].$$

It is evident that $\varphi[(D_{\boldsymbol{\theta}_{n+1}}^{\pi_{n+1}})^\top \boldsymbol{w}_{n+1}] > \mathbb{E}[(D^{\pi_\star})^\top \boldsymbol{w}_{n+1}]$ is not feasible. Thus we complete the proof. $\qquad\square$

### C.5 PROOF OF THEOREM 5

According to (Zahavy et al., 2021; Alegre et al., 2022b), for all $\boldsymbol{w}_{n+1} \in \boldsymbol{W}^\phi$, we can prove that

$$\mathbb{E}\left[(D^{\pi_\star})^\top \boldsymbol{w}_{n+1}\right] - \varphi\left[(D_{\boldsymbol{\theta}_{n+1}}^{\pi_{n+1}})^\top \boldsymbol{w}_{n+1}\right] \leq \mathbb{E}\left[(D^{\pi_\star})^\top \boldsymbol{w}_{n+1}\right] - v_{\boldsymbol{w}_{n+1}}^{\mathrm{SMP}} \leq \epsilon_1.$$

Therefore, for all $\boldsymbol{w}_{n+1} \in \boldsymbol{W}^\phi$, there exists an $\epsilon_2$ such that $\epsilon_2 \leq \epsilon_1$, let

$$\mathbb{E}\left[(D^{\pi_\star})^\top \boldsymbol{w}_{n+1}\right] - \varphi\left[(D_{\boldsymbol{\theta}_{n+1}}^{\pi_{n+1}})^\top \boldsymbol{w}_{n+1}\right] \leq \epsilon_2 \leq \epsilon_1. \tag{15}$$

By Theorem 3, for all $\boldsymbol{w}_{n+1} \in \boldsymbol{W}^\phi$, $(s,a) \in \mathcal{S} \times \mathcal{A}$, we have

$$\mathbb{E}\left[D^{\pi_\star}(s,a)^\top \boldsymbol{w}_{n+1}\right] - \varphi\left[D^{\pi_{n+1}}(s,a;\boldsymbol{\theta}_{n+1})^\top \boldsymbol{w}_{n+1}\right] \leq \delta_\varphi + \frac{2}{1-\gamma}\phi_{\max} \min_{j \in [n]} \|\boldsymbol{w}_{n+1} - \boldsymbol{w}_j\|. \tag{16}$$

According to (Alegre et al., 2022b), consider a new state space $\bar{\mathcal{S}} = \mathcal{S} \cup \{\bar{s}\}$, where $\bar{s}$ is a new dummy initial state in which only a single action $\bar{a}$ is available. Let $d_0(s) = p(s_0|\bar{s}, \bar{a})$ for all $s_0 \in \mu$, where $d_0(s_0)$ is the original probability of the initial state being $s_0$. Note that this does not change the values of any policy for the states $s \in \mathcal{S}$. Therefore,

$$\mathbb{E}\left[D^{\pi_\star}(\bar{s},\bar{a})^\top \boldsymbol{w}_{n+1}\right] - \varphi\left[D^{\pi_{n+1}}(\bar{s},\bar{a};\boldsymbol{\theta}_{n+1})^\top \boldsymbol{w}_{n+1}\right]$$
$$= \mathbb{E}\left[D^{\pi_\star}(\bar{s},\pi_\star(\bar{s}))^\top \boldsymbol{w}_{n+1}\right] - \varphi\left[D^{\pi_{n+1}}(\bar{s},\pi_{n+1}(\bar{s});\boldsymbol{\theta}_{n+1})^\top \boldsymbol{w}_{n+1}\right]$$
$$= \mathbb{E}[(D^{\pi_\star})^\top \boldsymbol{w}_{n+1}] - \varphi[(D_{\boldsymbol{\theta}_{n+1}}^{\pi_{n+1}})^\top \boldsymbol{w}_{n+1}]$$

Because Eq. (16) hold for $(\bar{s},\bar{a})$, for all $\boldsymbol{w}_{n+1} \in \boldsymbol{W}^\phi$ we have

$$\mathbb{E}[(D^{\pi_\star})^\top \boldsymbol{w}_{n+1}] - \varphi[(D_{\boldsymbol{\theta}_{n+1}}^{\pi_{n+1}})^\top \boldsymbol{w}_{n+1}] \leq \delta_\varphi + \frac{2}{1-\gamma}\phi_{\max} \min_{j \in [n]} \|\boldsymbol{w}_{n+1} - \boldsymbol{w}_j\|.$$

In the worst case, for $\boldsymbol{w}_{n+1} \in \boldsymbol{W}^\phi$, we have

$$\mathbb{E}[(D^{\pi_\star})^\top \boldsymbol{w}_{n+1}] - \varphi[(D_{\boldsymbol{\theta}_{n+1}}^{\pi_{n+1}})^\top \boldsymbol{w}_{n+1}] \leq \epsilon_2 \leq \delta_\varphi + \frac{2}{1-\gamma}\phi_{\max} \max_{\boldsymbol{w}_{n+1} \in \boldsymbol{W}^\phi} \min_{j \in [n]} \|\boldsymbol{w}_{n+1} - \boldsymbol{w}_j\|. \tag{17}$$

Combining Eq. (15) and Eq. (17), we complete the proof.

## D ADDITIONAL IMPLEMENTATION DETAILS OF RDSFOLS

In this section, we first exhibit an overall framework of RDSFOLS in Section D.1. Algorithm 1 in Section 4 is a solver of learning DSFs with DGPI. Two additional modules (corner weights and estimate improvement) are shown in Section D.2 and Section D.3, respectively.

### D.1 CONSTRUCTING A SET OF POLICIES WITH OPTIMISTIC LINEAR SUPPORT

Algorithm 2 takes an outer loop approach in which the set of CCS is incrementally constructed by learning DSFs (each DSFs corresponding to a task/MOMDP). It starts by inserting into a priority queue, $Q$, the weights in the extrema of the weight simplex $\boldsymbol{W}_e$ (i.e., weights in which one component is 1 and all others are 0), assigning them the maximum priority. For each iteration, we learn the weight $\boldsymbol{w}$ with the largest priority and apply Algorithm 1 to learn a policy $\pi$ and DSFs $D^\pi$ for

solving task $\boldsymbol{w}$. Following this, we recalculate corner weights and identify new weight vectors to add to $Q$. Detailed corner weight computation is provided within Algorithm 3. Algorithm 2 halts when either the priority queue $Q$ becomes empty or the maximum number of iterations is reached.

---

**Algorithm 2** Risk-sensitive DSFs Optimistic Linear Support (RDSFOLS)

1: **Initialize:** a set of policies $\Pi$ and the corresponding DSFs set $\mathcal{D}$; list of explored corner weights $\boldsymbol{W}$; priority queue $Q$
2: **for** each extremum of the weight simplex $\boldsymbol{w}_e \in \boldsymbol{W}_e$ **do**
3:      Add $\boldsymbol{w}_e$ to $Q$ with maximum priority
4: **end for**
5: **repeat**
6:      $\boldsymbol{w} \leftarrow$ pop weight with maximum priority in $Q$
7:      Update $\pi, D_{\boldsymbol{\theta}}^{\pi}$ by Alg. 1
8:      $\boldsymbol{W} \leftarrow \boldsymbol{W} \cup \boldsymbol{w}$
9:      **if** $D_{\boldsymbol{\theta}}^{\pi} \notin \mathcal{D}$ **then**
10:          Remove from $Q$ all $\boldsymbol{w}'$ s.t. $\mathbb{E}\left[(D_{\boldsymbol{\theta}}^{\pi})^{\top}\boldsymbol{w}'\right] \geq v_{\boldsymbol{w}'}^{\text{SMP}}$
11:          Apply Alg. 3 to obtain $\boldsymbol{W}_c$: $\boldsymbol{W}_c \leftarrow \text{CornerWeights}(D_{\boldsymbol{\theta}}^{\pi}, \boldsymbol{w}, \mathcal{D})$
12:          $\mathcal{D} \leftarrow \mathcal{D} \cup D_{\boldsymbol{\theta}}^{\pi}, \Pi \leftarrow \Pi \cup \pi$
13:          **if** $\boldsymbol{w}' \in \boldsymbol{W}_c$ **then**
14:              Apply Alg. 4 to get $\Delta(\boldsymbol{w}')$: $\Delta(\boldsymbol{w}') \leftarrow \text{EstimateImprovement}(\boldsymbol{w}', \mathcal{D}, \boldsymbol{W})$
15:              $Q \leftarrow Q \cup \Delta(\boldsymbol{w}')$
16:          **end if**
17:      **end if**
18: **until** $Q$ is empty
19: **return** $\Pi, \mathcal{D}$

---

### D.2 CORNER WEIGHTS

The task that RDSFOLS selects is determined by the corner weights in each iteration.

**Definition 4 (Corner weights (Roijers, 2016))** *Let $\mathcal{D}$ be a set of DSFs of policy set $\Pi$. Define the polyhedral subspace $P = \{\mathbf{x} \in \mathbb{R}^{d+1} | \mathbf{D}^{+}\mathbf{x} \leq \mathbf{0}, \forall k, w_k \geq 0, \sum_k w_k = 1\}$, where $\mathbf{D}^{+}$ is a matrix with the elements of $\mathcal{D}$ as row vectors, augmented by a column vector of $-1$'s. The vector $\mathbf{x} = (w_1, \ldots, w_d, \varphi\left[(D_{\boldsymbol{\theta}}^{\pi})^{\top}\boldsymbol{w}\right])$ consists of a weight vector and a risk value at those weights. The corner weights are the weights contained in the vertices of P.*

Intuitively, corner weights are the points where the piecewise-linear and convex (PWLC) surface changes slope. Here, PWLC represents a curve depicting the value of the SMP policy $v_{\boldsymbol{w}}^{\text{SMP}} = \max_{\pi \in \Pi} \varphi\left[(D_{\boldsymbol{\theta}}^{\pi})^{\top}\boldsymbol{w}\right]$, as a function of the task $\boldsymbol{w}$.

After calculating the new corner weights $\boldsymbol{W}_c$ at line 11 (in Algorithm 2), $D_{\boldsymbol{\theta}}^{\pi}$ is added to $\mathcal{D}$ at line 12. We now claim the reason why RDSFOLS can safely consider only corner weights, which is guaranteed by the following theorem.

**Theorem 6** *(Cheng, 1988) The maximum value of:*

$$\max_{\boldsymbol{w} \in \boldsymbol{W}^{\phi}, D_{\boldsymbol{\theta}}^{\pi} \in \text{CCS}} \min_{\pi' \in \Pi} \varphi[(D_{\boldsymbol{\theta}}^{\pi})^{\top}\boldsymbol{w}] - \varphi[(D_{\boldsymbol{\theta}'}^{\pi'})^{\top}\boldsymbol{w}] = \max_{\boldsymbol{w} \in \boldsymbol{W}^{\phi}} \varphi[(D^{\pi_\star})^{\top}\boldsymbol{w}] - v_{\boldsymbol{w}}^{\text{SMP}}$$

*is at one of the corner weights of $v_{\boldsymbol{w}}^{\text{SMP}} = \max_{\pi \in \Pi} \varphi[(D_{\boldsymbol{\theta}}^{\pi})^{\top}\boldsymbol{w}]$.*

Corner weights represent tasks for which the SMP policy yields values that deviate the most from their optimal values, i.e., they are the tasks that the agent knows the least about. Theorem 6 guarantees the correctness of RDSFOLS: once all corner weights have been examined, and no new DSFs are found, the maximal improvement must be 0, then Algorithm 2 has identified the complete CCS. Algorithm 3 shows the pseudocode for Computing the corner weights.

---

**Algorithm 3** Corner Weights (Roijers, 2016; Alegre et al., 2022b)

---

1: **Input:** New distributional SF vector $D_{\boldsymbol{\theta}}^{\pi}$, current weight vector $\boldsymbol{w}$, current DSFs set $\mathcal{D}$.
2: Let $\boldsymbol{W}_{del}$ be the set of obsolete weights removed from $Q$ in line 10 of Alg. 2
3: $\boldsymbol{W}_{del} \leftarrow \boldsymbol{W}_{del} \cup \boldsymbol{w}$
4: $\boldsymbol{V}_{rel} \leftarrow \{D_{\boldsymbol{\theta}}^{\pi} | D_{\boldsymbol{\theta}}^{\pi} \in \arg\max_{D_{\boldsymbol{\theta}}^{\pi} \in \mathcal{D}} \varphi[(D_{\boldsymbol{\theta}}^{\pi})^{\top} \boldsymbol{w}']$ for at least one $\boldsymbol{w}' \in \boldsymbol{W}_{del}\}$
5: $\boldsymbol{B}_{rel} \leftarrow$ the set of boundaries of the weight simplex $\boldsymbol{W}_e$ involved in any $\boldsymbol{w}' \in \boldsymbol{W}_{del}$
6: $\boldsymbol{W}_c \leftarrow \{\}$
7: **for** each subset $\mathcal{X}$ of $d-1$ elements from $\boldsymbol{V}_{rel} \cup \boldsymbol{B}_{rel}$ **do**
8:     $\boldsymbol{w}_c \leftarrow$ the weight in $\boldsymbol{W}_e$ where $D_{\boldsymbol{\theta}}^{\pi}$ intersects with the vectors/boundaries in $\mathcal{X}$
9:     Add $\boldsymbol{w}_c$ to $\boldsymbol{W}_c$
10: **end for**
11: **return** $\boldsymbol{W}_c$

---

### D.3 OPTIMISTIC MAXIMAL IMPROVEMENT

In general, there are many corner weights (candidate tasks to learn next). To determine the priority of tasks learning, a simple heuristic for exploring these tasks is an optimistic maximal improvement by calculating the difference between $v_{\boldsymbol{w}}^{\text{SMP}}$ and an optimistic upper bound on that task's optimal value $\bar{v}_{\boldsymbol{w}}^{\star}$. Notice that this heuristic only changes the order in which corner weights are explored, and this method does not impact the optimality of RDSFOLS.

---

**Algorithm 4** Estimate Improvement (Alegre et al., 2022b)

---

1: **Input:** New weight vector $\boldsymbol{w}$, DSFs set $\mathcal{D}$, set of weights, $\boldsymbol{W}$, for which optimal policies are already known.
2: Let $\bar{v}_{\boldsymbol{w}}^{\star}$ be the optimistic upper bound on $v_{\boldsymbol{w}}^{\star}$, computed by the following linear program (Diamond & Boyd, 2016):

$$\max \varphi[D^{\top} \boldsymbol{w}]$$
$$\text{s.t. } \varphi[D^{\top} \boldsymbol{w}'] \leq v_{\boldsymbol{w}'}^{\text{SMP}}, \text{ for all } \boldsymbol{w}' \in \boldsymbol{W}$$

3: $\Delta(\boldsymbol{w}) \leftarrow \bar{v}_{\boldsymbol{w}}^{\star} - v_{\boldsymbol{w}}^{\text{SMP}}$
4: **return** $\Delta(\boldsymbol{w})$

---

## E ADDITIONAL IMPLEMENTATION DETAILS OF DGPI-WCPI

In this subsection, we introduce the DGPI-WCPI algorithm, which is the control group of WCPI; as shown in Algorithm 5. Different from WCPI, DGPI-WCPI is designed to learn a diverse set of policies, such that the performance of the DGPI defined on that policy set will have the optimal worst-case performance among all tasks $\boldsymbol{W}^{\phi}$. For details on how to compute the worst-case reward (line 5) by solving linear programs, see (Zahavy et al., 2021, Lemma 4).

---

**Algorithm 5** DGPI Worst Case Distributional Policy Iteration

---

1: **Input:** a set of policies $\Pi$ and correspond DSFs set $\mathcal{D}$; sample weight vector $\bar{\boldsymbol{w}} \sim \boldsymbol{W}^{\phi}$
2: For task $\bar{\boldsymbol{w}}$, update $\pi, D_{\boldsymbol{\theta}}^{\pi}$ by Alg. 1
3: $\mathcal{D} \leftarrow \mathcal{D} \cup D_{\boldsymbol{\theta}}^{\pi}, \Pi \leftarrow \Pi \cup \pi$
4: **repeat**
5:     $\bar{\boldsymbol{w}} \leftarrow \arg\min_{\boldsymbol{w} \in \boldsymbol{W}^{\phi}} \max_{\pi \in \Pi} \mathbb{E}[(D_{\boldsymbol{\theta}}^{\pi})^{\top} \boldsymbol{w}]$
6:     For task $\bar{\boldsymbol{w}}$, update $\pi, D_{\boldsymbol{\theta}}^{\pi}$ by Alg. 1
7:     $\mathcal{D} \leftarrow \mathcal{D} \cup D_{\boldsymbol{\theta}}^{\pi}, \Pi \leftarrow \Pi \cup \pi$
8: **until** $\varphi[(D_{\boldsymbol{\theta}}^{\pi})^{\top} \bar{\boldsymbol{w}}]$ does not improve
9: **return** $\Pi, \mathcal{D}$

---

# F    EXPERIMENT DETAILS

## F.1    BENCHMARK ENVIRONMENTS

For a comprehensive evaluation, we have designed two new MORL environments with discrete action space based on MuJoCo (Todorov et al., 2012), named DiscreteHopper and DiscreteSwimmer, except for classic multi-task environments like Reacher. The detailed explanation of the three environments is as follows.

**Reacher** is a modification of "Reacher-v4" from PyBullet (Alegre et al., 2022a), which is a well-known benchmark environment used in the SFs literature (Barreto et al., 2017; Gimelfarb et al., 2021; Nemecek & Parr, 2021). The agent operates a dual-segmented arm by applying torque to the two joints, and reward features $\phi(s, a, s') \in \mathbb{R}^4$ are defined as one minus the Euclidean distance from the tip of the robotic arm to the four target position.

**DiscreteHopper** is a modification of "Hopper-v4". The three-dimensional action space $\mathcal{A} \subset [-1, +1]^3$ is uniformly discretized using 21 values. The agent is a single-legged hopper, a two-dimensional figure with four body parts, and the aim is to move it forward by applying torque to its three hinges. We follow the reward features setting from the MO-Gym library (Alegre et al., 2022a), which encode three different dimensional reward functions: x-axis going forward, z-axis jumping high and action cost.

**DiscreteSwimmer** is a modification of "Swimmer-v3". The two-dimensional action space $\mathcal{A} \subset [-1, +1]^2$ is discretized using seven uniform values per dimension. The swimmers aim to move efficiently to the right in a 2D pool by using rotor torque and fluid friction. The reward features are composed of six components. The first component "forward reward" and the second component "control cost" continue to use the composition of the original reward. The third and fourth components describe the changes of the position in the x-axis and y-axis, respectively. The last two components depict the x-axis velocity and the y-axis velocity.

## F.2    QUANTILE FRACTION GENERATION

For approximating the distribution SFs, we adopt quantile regression to learn quantile values. In our work, fix (QR-DQN), random (IQN) and net (FQF) are employed to generate quantile fractions. In particular, the quantile proposal network in FQF is designed as a two-layer fully connected network, with each layer containing 128 units. Additionally, it operates with a learning rate of 1e-5 (Ma et al., 2020). In Fig. 4, we provide a modular experiment and select the random method for quantile fraction generation, as it outperforms fix and has fewer parameters than net, based on experiment results.

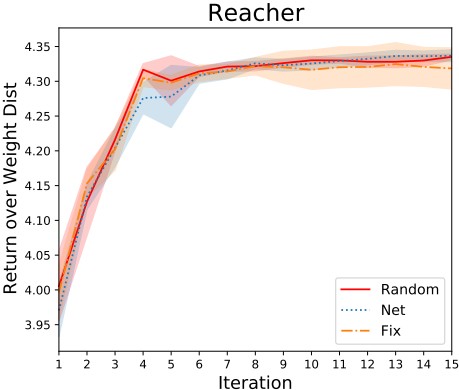

Figure 4: Comparison of quantile fraction generation methods in Reacher. Each learning curve is averaged over five different random seeds.

## F.3 RISK OPERATORS

In this subsection, except for the risk-neutral measure function, we elaborate on the additional five risk operators mentioned in Section 3.2.

- For mean-variance (Sobel, 1982; Tamar et al., 2012; Prashanth & Ghavamzadeh, 2016), $\varphi(Z) = \mathbb{E}[Z] - \beta\sqrt{\mathbb{V}[Z]}$, where $\sqrt{\mathbb{V}[Z]}$ is the standard deviation. $\beta$ is set as 0.1 for risk-averse, while -0.1 for risk-seeking.
- VaR (Prashanth L & Fu, 2018) quantifies risk as the minimum reward (maximum cost) that could occur at a specified confidence level (Chow et al., 2015; L.A. & Fu, 2018; Ma et al., 2020). Its explicit expression is $\mathrm{VaR}_\beta(Z) = \min_z \{z | F_Z(z) > \beta\}$, where $\beta$ is taken as 0.25 for risk-averse and $\beta$ is taken as 0.75 for risk-seeking.

The remaining three risk operators are distorted expectations. Specifically, let $g(\tau)$ be the distortion function.

- For Wang's method (Wang, 2000), $g(\tau) = \Phi(\Phi^{-1}(\tau) + \beta)$, where $\Phi$ and $\Phi^{-1}$ are the standard Normal CDF and its inverse. $\beta$ is chosen as 0.75 for risk-averse, while -0.75 for risk-seeking.
- The risk operator CPW (Tversky & Kahneman, 1992) is only suitable for risk-averse. Its detailed expression is $g(\tau) = \tau^\beta / \left(\tau^\beta + (1 - \tau)^\beta\right)^{1/\beta}$, where $\beta$ is set as 0.71 for most closely human subjects (Ma et al., 2020).
- CVaR (Chow et al., 2015) is also a risk-averse method, where $g(\tau) = \min\{\tau/\beta, 1\}$, and $\beta$ is usually taken as 0.25 for available.

## F.4 POLICY TRAINING UNDER RISK OPERATORS

We compare different risk operators on benchmark environments:

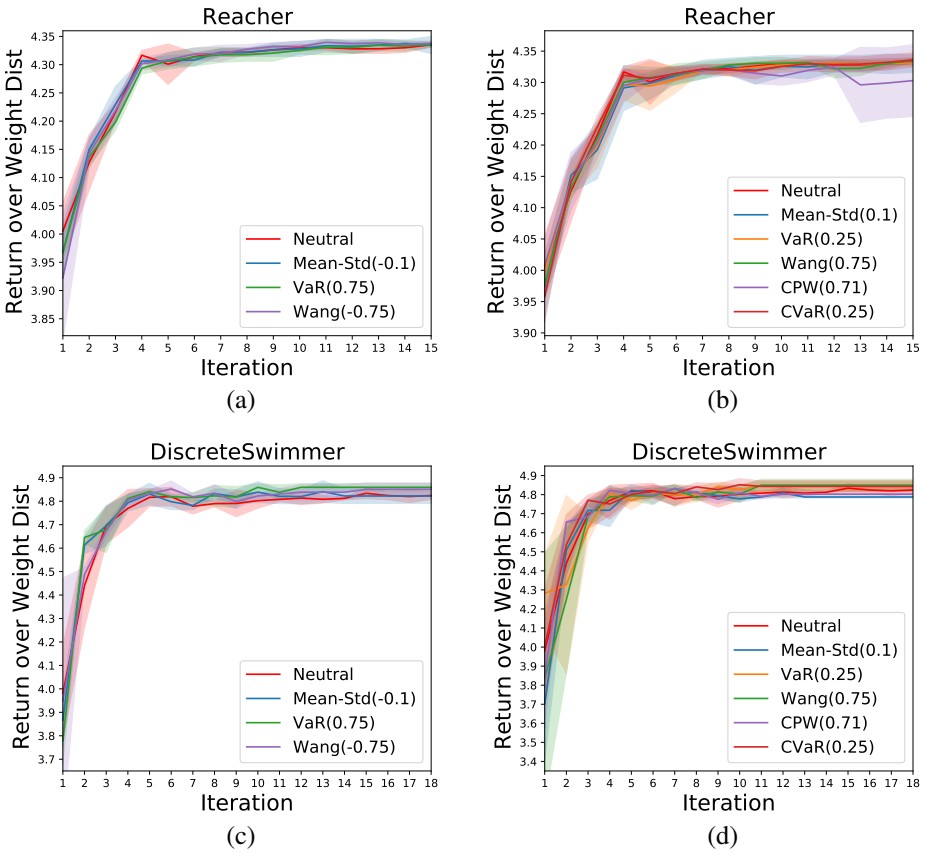

(a)

(b)

(c)

(d)

Figure 5: Comparison of the RDSFOLS algorithm using different risk operators. Each curve is run with five seeds. (a) risk-seeking policies in Reacher, (b) risk-averse policies in Reacher. (c) risk-seeking policies in DiscreteSwimmer, (d) risk-averse policies in DiscreteSwimmer.

