# OpenReview forum: "Tackling Underestimation Bias in Successor Features by Distributional Reinforcement Learning"
_ICLR.cc/2024/Conference — ICLR 2024 Conference Withdrawn Submission_

### Official Review · Reviewer_mAvf · 2023-10-31

**Soundness:** 1 poor
**Presentation:** 1 poor
**Contribution:** 2 fair
**Rating:** 3
**Confidence:** 3

**Summary:**

In this paper, the authors identify an issue with underestimation bias when employing successor features with generalized policy improvement (GPI). They apply distributional RL to successor features to obtain a distributional GPI that they claim alleviates this issue.

**Strengths:**

* Connecting successor features and multi-objective RL. These two concepts are intimately related, yet I’m unaware of any published work making a direct connection.
* Identifying the issue of underestimation with SFs. The intuition behind this makes sense but I found the explanation hard to follow.

**Weaknesses:**

Overall, I found the paper hard to follow as the math is a little too loose to make sense of what the authors are claiming. I'd suggest the authors go over the text with a careful eye and fix some of the notational issues. I’ll detail the major weaknesses and ask more direct questions about notation and other issues in the questions section below.

* I believe they are learning the wrong object with their proposed distributional SF algorithm. They should be learning the features' joint distribution, but they treat each feature as independent. This is implicitly done in how the author's sample $\tau$; to learn the proper distribution, you'd need the quantile of a multivariate random variable.
* The paper is poorly positioned in the literature regarding multi-dimensional reward functions in distributional RL. My above point on learning the wrong object has been solved by [1] and [2], where they learn the correct joint distribution. Furthermore, I expected a more in-depth discussion about [3] as they also learn “distributional SFs” (although they also aren’t learning the joint distribution).
* Assumption 1 seems dubious; this should be impossible with stochastic transitions; a single application of the Bellman operator will construct a mixture distribution, so, at the very least, you’d expect the target to be a mixture of Gaussians.
* No justification is given for the additive noise model (Equation 6).
* It’s hard to judge the effectiveness of the approach as the empirical results don’t differentiate much between distributional GPI and regular GPI. I would have had to see a better-executed empirical study to be convinced.

---

[1] Pushi Zhang, Xiaoyu Chen, Li Zhao, Wei Xiong, Tao Qin, Tie-Yan Liu. Distributional Reinforcement Learning for Multi-Dimensional Reward Functions. NeurIPS 2021.

[2]  Dror Freirich, Tzahi Shimkin, Ron Meir, Aviv Tamar. Distributional Multivariate Policy Evaluation and Exploration with the Bellman GAN. ICML 2019.

[3] Michael Gimelfarb, Andre Barreto, Scott Sander, and Chi-Guhn Lee. Risk-Aware Transfer in Reinforcement Learning using Successor Features. NeurIPS 2021.

**Questions:**

- In Section 3.1, the prediction $\Psi(s', a', \theta_i)$ should be $\Psi(s, a, \theta_i)$? Only the TD target should have the next state-action term. This error is propagated from this point forward, e.g., the gradient term is wrong, and eq (5) contains the same error.
- In Section 3.1, why is there an expectation over s’ in the loss? Aren’t we trying to write down the stochastic approximation algorithm for learning SFs via TD? Citing equation 1 makes it seem like that’s what we’re trying to do.
- Theorem 1, why is $s’$, $a’$ defined as input to $\Delta$ but then $s'$ appears in the expectation? Also, why do we have an expectation over $s'$ again?
- Theorem 3 compares the expected return of the optimal policy with a risk measure of the estimated policy. Why? If you're being risk-sensitive, the goal is not to learn the mean-optimal policy.
- Algorithm 1, where is $\tau_e$ being used? Shouldn't it be used in Line 6 when computing the greedy action?
- Algorithm 1, quantiles are treated implicitly in some cases; this makes it hard to decipher what's going on.

---

> ### Author Response · Authors · 2023-11-20
> **Answer to Reviewer mAvf 1**
>
> 1.*``I believe they are learning the wrong object with their proposed distributional SF algorithm. They should be learning the features' joint distribution, but they treat each feature as independent. This is implicitly done in how the author's sample $\tau$; to learn the proper distribution, you'd need the quantile of a multivariate random variable. ’’*
>
> 2.*``The paper is poorly positioned in the literature regarding multi-dimensional reward functions in distributional RL. My above point on learning the wrong object has been solved by [1] and [2], where they learn the correct joint distribution.’’*
>
> 4.*``Assumption 1 seems dubious; this should be impossible with stochastic transitions; a single application of the Bellman operator will construct a mixture distribution, so, at the very least, you’d expect the target to be a mixture of Gaussians.’’*
>
> **Answer.** Thank you for your valuable suggestion. As in [3], we will consider a more general case elliptical distribution.
>
> 3.*``Furthermore, I expected a more in-depth discussion about [3] as they also learn “distributional SFs” (although they also aren’t learning the joint distribution).’’*
>
> **Answer.** Thank you very much for the insightful references. We will add [3] to the related work section. Specifically, Gimelfarb et al. proposed risk-aware successor features (RaSF) to realize policy transfer between tasks with shared dynamics by incorporating risk-awareness into the decision-making. Differently, our proposed concept of DSFs considers multiple different risk measures, which consider more scenarios of the return distribution and are more general.
>
>
> [1] Zhang et al.. Distributional reinforcement learning for multi-dimensional reward functions. NeurIPS 2021.
>
> [2] Freirich et al.. Distributional multivariate policy evaluation and exploration with the bellman GAN. ICML 2019.
>
> [3] Gimelfarb et al.. Risk-aware transfer in reinforcement learning using successor features. NeurIPS 2021.

---

> ### Author Response · Authors · 2023-11-20
> **Answer to Reviewer mAvf 2**
>
> 5.*``No justification is given for the additive noise model (Equation 6).’’*
>
> **Answer.** Actually, we have claimed the justification of the additive noise model before Eq. (6). Specifically, in practical applications, SFs-estimation usually incorporates random errors, which presumably stem from function approximation and are induced by source tasks.
>
> 9.*``Theorem 1, why is $s′, a′$ defined as input to $\Delta$ but then $s′$ appears in the expectation? Also, why do we have an expectation over $s′$ again?’’*
>
> **Answer.** Thank you for your valuable suggestion. We approve that $\Delta$ is not a function of $s’,a’$, and we will correct it in our further revision.
>
> 10.*``Theorem 3 compares the expected return of the optimal policy with a risk measure of the estimated policy. Why? If you're being risk-sensitive, the goal is not to learn the mean-optimal policy. ’’*
>
> **Answer.** Actually, the estimated policy is trained by risk-sensitive RL algorithms,  while the optimal policy is determined by the MDP independent with aleatoric risk measures. Thus, the estimated policy and the optimal policy should be considered in different scenarios, where the former (inherent) is objective and the latter (trained) is risk-sensitive.
>
> 11.*``Algorithm 1, where is $\tau_{e}$ being used? Shouldn't it be used in Line 6 when computing the greedy action?’’*
>
> 12.*``Algorithm 1, quantiles are treated implicitly in some cases; this makes it hard to decipher what's going on. ’’*
>
> **Answer.** Thank you for your careful review. We will elaborate on how to use various kinds of quantiles in our further revision. Specifically, indeed, $\tau_{e}$ is used in Line 6 when computing the greedy action. $\tau_{j}$ and $\tau_{k}$ are used to compute the loss $J_{Z}(\boldsymbol{\theta})$. All these issues will be clarified.

---

> > ### Comment · Reviewer_mAvf · 2023-11-21
> >
> > I believe my main concerns were still not addressed in the rebuttal, specifically points (1) and (2) above. Furthermore, my confusion around comparing the optimal policy w.r.t. the expected return and the optimal policy w.r.t. a risk measure still remains. That said, I'd still like to keep my score as is, I believe it's too late to adequately address the issues with this paper for ICLR.

---

> ### Author Response · Authors · 2023-11-21
>
> We appreciate the time and effort that you dedicated to providing feedback on our manuscript and are grateful for the insightful comments on our paper.

---

### Official Review · Reviewer_3ktK · 2023-11-04

**Soundness:** 2 fair
**Presentation:** 1 poor
**Contribution:** 1 poor
**Rating:** 3
**Confidence:** 2

**Summary:**

In this work, the authors aim to address underestimation when using Successor Features and Generalized Policy Iteration (GPI). A common technique often used to prevent overestimation in the Q-values is by using the min operation with double Q-functions, which may in turn result in underestimation. Motivated by this insight,  the authors rely on theoretical analyses to show a similar trait is observed when updating the parameters of the successor features. This is induced by a mismatch of the parameters of the successor features between one that depends on a changing policy distribution and the other being the optimal policy. The authors proposed replacing successor features with its distributional form in order to limit or reduce the underestimation bias.

**Strengths:**

I have to be honest. It is difficult for me to identify what to write with regards to the strength of the paper. I can see that a lot of work has been done. However, the presentation and writing is not doing justice to the authors’ effort.

**Weaknesses:**

1. First of all, the writing and the overall presentation is not very clear and does not flow well at times. Despite reading the paper a couple of times, it still seems confusing and overly complicated. Lastly, some of the sentences in the paper do not even make sense and makes me wonder if they were generated by LLMs. Here are some examples:
  a. “Explosively, we take an impressive TRL method - successor features (SFs) (Barreto et al., 2017; 2018; Carvalho et al., 2023) as an example, to study the underlying overestimation/underestimation bias.”
  b.“They enrich our concepts mutually.”
  c.“Extensive quantitative evaluations support our analysis.”
2. It seems that the research question about addressing underestimation was motivated by RL/ But at the moment, I fail to understand the need of using risk-sensitive frameworks and multi-objective RL. What is the main motivation for considering these frameworks and theories? Furthermore, the lack of clarity from the section on bridging successor features and multi-objective RL does not help the cause.
3. Eq 4. Is y the target that you are regressing towards? It is confusing if that is not the case.
4. It is very hard to read the paper when there are a large portion of different concepts and their corresponding theorems and equations. I would recommend moving most of these items into the appendix and use the main portion of the paper to explain what these different concepts are and how they are related to the research question that you are attempting to address. You can also move the pseudocode for Algorithm 1 into the appendix. This will also allow you to make more space for the section for conclusion and discussion.
5. The overall paper structure should be re-visited. The fact that a whole chunk of related work is in your appendix is a missed opportunity for the readers that they can follow along.
6. Although the author did provide the theoretical proofs showing the existence of the underestimation bias in the SF & GPI framework, this point will make a stronger case with empirical evidence as well.

**Questions:**

1. What is the purpose of analyzing using the risk-sensitive framework?
2. What is the purpose of considering multi-objective RL which only further complicates the study?

---

> ### Author Response · Authors · 2023-11-20
> **Answer to Reviewer 3ktK 1**
>
> 1.*``First of all, the writing and the overall presentation is not very clear and does not flow well at times. Despite reading the paper a couple of times, it still seems confusing and overly complicated. Lastly, some of the sentences in the paper do not even make sense and makes me wonder if they were generated by LLMs. Here are some examples: a. “Explosively, we take an impressive TRL method - successor features (SFs) (Barreto et al., 2017; 2018; Carvalho et al., 2023) as an example, to study the underlying overestimation/underestimation bias.” b.“They enrich our concepts mutually.” c.“Extensive quantitative evaluations support our analysis.”’’*
>
> **Answer.** Thank you for your careful review. Actually, we do not use LLMs while writing the paper, and we are sorry for all these unclear expressions. We will update these sentences in our further revision. The specifics are as follows.
>
> Explosively, we take an impressive TRL method - successor features (SFs) (Barreto et al., 2017; 2018; Carvalho et al., 2023) as an example, to study the underlying overestimation/underestimation bias. -> Typically, we study the underlying overestimation/underestimation bias in TRL by taking a representative method - successor features (SFs) (Barreto et al., 2017; 2018; Carvalho et al., 2023) as an example.
>
> They enrich our concepts mutually -> We will delete this meaningless sentence in our further revision.
>
> Extensive quantitative evaluations support our analysis -> Extensive quantitative evaluations demonstrate the Q-value underestimation phenomenon in SFs transfer framework and the effect of DSFs mitigating such underestimation.
>
> 2.*``It seems that the research question about addressing underestimation was motivated by RL/ But at the moment, I fail to understand the need of using risk-sensitive frameworks and multi-objective RL. What is the main motivation for considering these frameworks and theories? Furthermore, the lack of clarity from the section on bridging successor features and multi-objective RL does not help the cause.’’
>
> 7.``What is the purpose of analyzing using the risk-sensitive framework?’’
>
> 8.``What is the purpose of considering multi-objective RL which only further complicates the study?’’*
>
> **Answer.**
>
> **The main motivation for risk-sensitive frameworks.**
>
> As mentioned in [1], ‘risk’ refers to uncertainty over possible outcomes, and risk-sensitive policies are those that depend upon more than the mean of the outcomes. Meanwhile, [2] claims that using the expected return as a measure of optimality could still lead to undesirable behavior such as excessive risk-taking, since low-probability catastrophic outcomes with negative reward and high variance could be underrepresented. For this reason, risk awareness is becoming an important aspect of practical RL. With the aid of the risk-sensitive framework, a series of RL works have been successfully conducted [3, 4]. In our paper, we leverage this framework to relieve the underestimation problem in the scenario of SFs.
>
> **The main motivation for MORL.**
>
> Incorporating MORL will mitigate the underestimation, and subsequently ensure possessing the optimal policy. Specifically, GPI/DGPI can only be used to construct a policy that performs no worse than the existing one but brings about the limitation of the policies' optimality. The prior knowledge of the set of SFs/DSFs is crucial for the precise and efficient learning of optimal policies using GPI/DGPI. To tackle this issue, MORL literature has extensively studied the problem of constructing an excellent prior, convex coverage set (CCS). For detailed literature, please refer to [5, 6].
>
> Actually, we should also conduct experiments on separate SFs. However, due to program limitations, we only conduct experiments by combining SF with MORL. We aim to address this limitation in the future and include experiments on separate SFs to improve our paper.
>
> **The lack of clarity from the section on bridging successor features and multi-objective RL does not help the cause.**
>
> Thank you for your valuable suggestion. We will complement the purpose of bridging SFs and MORL in Section 2.2 of our revision.
>
>
> [1] Dabney et al.. Implicit quantile networks for distributional reinforcement learning. ICML, 2018.
>
> [2] Gimelfarb et al.. Risk-aware transfer in reinforcement learning using successor features. NeurIPS 2021.
>
> [3] Ma et al.. DSAC: Distributional soft actor critic for risk-sensitive reinforcement learning. arXiv preprint arXiv: 2004.14547.2020.
>
> [4] Zhou et al.. Distributional generative adversarial imitation learning with reproducing kernel generalization. Neural Networks, 2023.
>
> [5] Yang et al.. A generalized algorithm for multi-objective reinforcement learning and policy adaptation. NeurIPS 2019.
>
> [6] Hayes et al.. A practical guide to multi-objective reinforcement learning and planning. AAMAS, 2022.

---

> > ### Comment · Reviewer_3ktK · 2023-11-21
> >
> > I thank the authors for the response to my questions and concerns, as well as sharing the resources on bridging SFs and multi-objective RL. Given that there is still substantial amount of work to be done, I will keep my score as it is. I hope that the authors will incorporate the suggestions that other reviewers and myself have provided in order to improve the quality of the paper.

---

> ### Author Response · Authors · 2023-11-20
> **Answer to Reviewer 3ktK 2**
>
> 4.*``It is very hard to read the paper when there are a large portion of different concepts and their corresponding theorems and equations. I would recommend moving most of these items into the appendix and use the main portion of the paper to explain what these different concepts are and how they are related to the research question that you are attempting to address. You can also move the pseudocode for Algorithm 1 into the appendix. This will also allow you to make more space for the section for conclusion and discussion. ’’*
>
> 5.*``The overall paper structure should be re-visited. The fact that a whole chunk of related work is in your appendix is a missed opportunity for the readers that they can follow along.’’*
>
> **Answer.** Thank you for your valuable suggestion. We will reorganize the whole paper for a more clarified presentation.

---

> ### Author Response · Authors · 2023-11-21
>
> We appreciate the time and effort that you dedicated to providing feedback on our manuscript and are grateful for the insightful comments on our paper.

---

### Official Review · Reviewer_dNHm · 2023-11-05

**Soundness:** 3 good
**Presentation:** 1 poor
**Contribution:** 2 fair
**Rating:** 5
**Confidence:** 3

**Summary:**

This paper theoretically studies the underestimation phenomenon in successor features and generalized policy improvement. The paper introduces distributional RL into the SF/GPI framework so as to mitigate underestimation and theoretically analyzes its generalization bounds. The experiments are run on multi-objective RL environments (testing transferabiity with GPI) in Mujoco. They compare the performance of their distributional variants to the standard SF/GPI variants.

**Strengths:**

The paper does indeed seem to show theoretically that the underestimation phenomena occurs in SF & GPI.

The paper does indeed seem to show that distributional SFs has a lower generalization bound than the original SFs.

**Weaknesses:**

My primary concerns have to do with clarity, as well as well as the experimental results.

There are many typos or awkward wording, including some that impact the reader's understanding.

Some examples include:
- "The results indicate that both the two DSFs-based algorithms (RDSFOLS and DGPI-WCPI)". The latter isn't even in Figure 2? I am assuming the latter refers to "WCDPI+DGPI", but this should be clarified.
Awkward wording:
- "resulting in a “zero-shot” somewhat fantastical"
- "For a new task w_{n+1}, it is practicable to evaluate all policies". Perhaps you mean practical?
- "standpoint to expose the mystery of underestimation". The word 'mystery' seems akward here.
- "Due to the disorder of exploration": I don't know what "disorder of explanation refers to"
- "is lack of stability"
- "DSFs exhibits" -> "exhibit"
- "We remark that δ_φ > 0 makes no focus". I didn't understand what was meant by "focus".
- "if the set of DSFs enough close"

There are many more beyond what was mentioned, and this does indeed negatively impact the readability of the paper.

The results in Figure 2 do not appear to be very compelling. It seems the proposed method does not significantly outperform the baselines.

**Questions:**

- Can you describe again the y-axis in Figure 2?
- Did you look at Q-value predictions of the agents to show/demonstrate lower underestimation?

---

> ### Author Response · Authors · 2023-11-20
> **Answer to Reviewer dNHm**
>
> 1. *``The results indicate that both the two DSFs-based algorithms (RDSFOLS and DGPI-WCPI)". The latter isn't even in Figure 2? I am assuming the latter refers to "WCDPI+DGPI", but this should be clarified. ’’*
>
> **Answer.** Thank you for pointing this out. The label "WCDPI+DGPI" should be "DGPI-WCPI". We are sorry for this typo, and we will correct it in our revised paper.
>
> 2.*``Awkward wording.*
>
> *"resulting in a “zero-shot” somewhat fantastical"*
>
> *"For a new task $w_{n+1}$, it is practicable to evaluate all policies". Perhaps you mean practical?*
>
> *"standpoint to expose the mystery of underestimation". The word 'mystery' seems akward here.*
>
> *"Due to the disorder of exploration": I don't know what "disorder of explanation refers to"*
>
> *"is lack of stability"*
>
> *"DSFs exhibits" -> "exhibit"*
>
> *"We remark that $\delta_{\phi}>0$ makes no focus". I didn't understand what was meant by "focus".*
>
> *"if the set of DSFs enough close"’’*
>
> **Answer.** Thank you for your careful review and suggestions. We will make updates to these linguistic issues. The specifics are as follows.
>
> For a new task $w_{n+1}$, it is practicable to evaluate all policies -> For a new task $w_{n+1}$, it is practical to evaluate all policies
>
> standpoint to expose the mystery of underestimation -> standpoint to expose the phenomenon of underestimation
>
> Due to the disorder of exploration -> Due to the randomness of exploration
> is lack of stability -> is not stable
>
> DSFs exhibits -> exhibit, and we will correct this issue throughout the whole paper.
>
> The agent will attain near-optimal performance, if the set of DSFs enough close to task $w_{n+1}$.  -> The agent will attain near-optimal performance in task $w_{n+1}$, if it has solved a similar task before.
>
> 4.*``Can you describe again the y-axis in Figure 2?’’*
>
> **Answer.** In Figure 2, the y-axis represents the expected return (value) obtained from each method when evaluated over a test set of 60 tasks uniformly sampled from $\boldsymbol{W}^{\boldsymbol{\phi}}$.

---

### Author Response · Authors · 2023-11-20
**Tackling Underestimation Bias in Successor Features by Distributional Reinforcement Learning**

The authors express gratitude to the reviewers and chairs for their valuable comments. Below are our point-to-point replies to reviewers' comments.

---

> ### Author Response · Authors · 2023-11-20
> **Response to Reviewers' Common Problems**
>
> Reviewer dNHm-5.*``Did you look at Q-value predictions of the agents to show/demonstrate lower underestimation?’’*
>
> Reviewer 3ktK-6.*``Although the author did provide the theoretical proofs showing the existence of the underestimation bias in the SF & GPI framework, this point will make a stronger case with empirical evidence as well.’’*
>
> **Answer.** Actually, the dashed lines in Figure 2 denote the true Q-values in the new task, and the solid lines mean Q-value predictions of the agents. Obviously, Q-value predictions are lower than the true Q-values in the new task. This supports the theoretical analysis in Section 3.1.
>
> Reviewer 3ktK-3.*``Eq 4. Is y the target that you are regressing towards? It is confusing if that is not the case. ’’*
>
> Reviewer mAvf-7.*``In Section 3.1, the prediction $\Psi(s′,a′,\theta_{i})$ should be $\Psi(s,a,\theta_{i})$? Only the TD target should have the next state-action term. This error is propagated from this point forward, e.g., the gradient term is wrong, and eq (5) contains the same error. ’’*
>
> Reviewer mAvf-8.*``In Section 3.1, why is there an expectation over s’ in the loss? Aren’t we trying to write down the stochastic approximation algorithm for learning SFs via TD? Citing equation 1 makes it seem like that’s what we’re trying to do. ’’*
>
> **Answer.** We are sorry for this typo. Actually, term $y$-$\psi^{\pi_{n+1}}(s',a';\theta_{i})^{T}w_{n+1}$ should be $y$-$\psi^{\pi_{n+1}}(s,a;\theta_{i})^{T}w_{n+1}$, which denotes the TD error.
>
> On the other hand, owing to the fact that the updated parameter $\theta_{n+1}$, rather than $\boldsymbol{\theta}_{i}$, is included in the greedy target value with respect to $s’,a’$, the gradient term is subsequently with respect to $s’,a’$ and contains the expectation over $s’$.
>
> We will correct it in our further revision. Thanks a lot.
>
> Reviewer dNHm-3.*``The results in Figure 2 do not appear to be very compelling. It seems the proposed method does not significantly outperform the baselines.’’*
>
> Reviewer mAvf-6.*``It’s hard to judge the effectiveness of the approach as the empirical results don’t differentiate much between distributional GPI and regular GPI. I would have had to see a better-executed empirical study to be convinced. ’’*
>
> **Answer.** Thank you for your insightful comment. Actually, we should conduct experiments on separate SFs and SFs with MORL. However, due to program limitations, we only conduct the latter. The reason why the experiment is not obvious is mainly because the x-axis is the number of iterations, which is followed by [1]. It is convinced that there will be an obvious good performance in both sample efficiency and stability for DSFs under the measure of training steps. We will strive to complement the experiments on separate SFs in the future.
>
> [1] Alegre et al.. Optimistic linear support and successor features as a basis for optimal policy transfer. In ICML, pp. 394–413, 2022.